# The Complexity of Symmetric Equilibria in Min-Max Optimization and Team Zero-Sum Games[*]

Ioannis Anagnostides[1], Ioannis Panageas[2], Tuomas Sandholm[1,3], and Jingming Yan[2]

[1]Carnegie Mellon University
[2]University of California, Irvine
[3]Strategy Robot, Inc.
[3]Strategic Machine, Inc.
[3]Optimized Markets, Inc.
{ianagnos,sandholm}@cs.cmu.edu, {ipanagea,jingmy1}@uci.edu

## Abstract

We consider the problem of computing stationary points in min-max optimization, with a focus on the special case of Nash equilibria in (two-)team zero-sum games. We first show that computing $\epsilon$-Nash equilibria in 3-player *adversarial* team games—wherein a team of 2 players competes against a *single* adversary—is CLS-complete, resolving the complexity of Nash equilibria in such settings. Our proof proceeds by reducing from *symmetric* $\epsilon$-Nash equilibria in *symmetric*, identical-payoff, two-player games, by suitably leveraging the adversarial player so as to enforce symmetry—without disturbing the structure of the game. In particular, the class of instances we construct comprises solely polymatrix games, thereby also settling a question left open by Hollender, Maystre, and Nagarajan (2024).

Moreover, we establish that computing *symmetric* (first-order) equilibria in *symmetric* min-max optimization is PPAD-complete, even for quadratic functions. Building on this reduction, we show that computing symmetric $\epsilon$-Nash equilibria in symmetric, 6-player (3 vs. 3) team zero-sum games is also PPAD-complete, even for $\epsilon = \text{poly}(1/n)$. As a corollary, this precludes the existence of symmetric dynamics—which includes many of the algorithms considered in the literature—converging to stationary points. Finally, we prove that computing a *non-symmetric* $\text{poly}(1/n)$-equilibrium in symmetric min-max optimization is FNP-hard.

## 1 Introduction

We consider computing local equilibria in constrained min-max optimization problems of the form

$$\min_{\boldsymbol{x} \in \mathcal{X}} \max_{\boldsymbol{y} \in \mathcal{Y}} f(\boldsymbol{x}, \boldsymbol{y}), \tag{1}$$

where $\mathcal{X} \subseteq \mathbb{R}^{d_x}$ and $\mathcal{Y} \subseteq \mathbb{R}^{d_y}$ are convex and compact constraint sets, and $f : \mathcal{X} \times \mathcal{Y} \to \mathbb{R}$ is a smooth objective function. Tracing all the way back to Von Neumann's celebrated minimax theorem [von Neumann, 1928] and the inception of game theory, such problems are attracting renewed interest in recent years propelled by a variety of modern machine learning applications, such as generative modeling [Goodfellow et al., 2014], reinforcement learning [Daskalakis et al., 2020, Bai and Jin, 2020, Wei et al., 2021], and adversarial robustness [Madry et al., 2018, Cohen et al., 2019, Bai et al., 2021, Carlini et al., 2019]. Another prominent class of problems encompassed by (1) concerns computing Nash equilibria in *(two-)team zero-sum games* [Zhang et al., 2023, 2021, Basilico et al., 2017, von Stengel

---

[*]The authors are ordered alphabetically.

39th Conference on Neural Information Processing Systems (NeurIPS 2025).

and Koller, 1997, Carminati et al., 2023, Orzech and Rinard, 2023, Farina et al., 2018, Zhang and An, 2020, Celli and Gatti, 2018, Schulman and Vazirani, 2017], which is a primary focus of this paper.

Perhaps the most natural solution concept—guaranteed to always exist—pertaining to (1), when $f$ is nonconvex-nonconcave, is a pair of strategies $(\boldsymbol{x}^*, \boldsymbol{y}^*)$ such that both players (approximately) satisfy the associated *first-order* optimality conditions [Tsaknakis and Hong, 2021, Jordan et al., 2023, Ostrovskii et al., 2021, Nouiehed et al., 2019], as formalized in the definition below.

**Definition 1.1.** A point $(\boldsymbol{x}^*, \boldsymbol{y}^*) \in \mathcal{X} \times \mathcal{Y}$ is an $\epsilon$-*first-order Nash equilibrium* of (1) if

$$\langle \boldsymbol{x} - \boldsymbol{x}^*, \nabla_{\boldsymbol{x}} f(\boldsymbol{x}^*, \boldsymbol{y}^*) \rangle \geq -\epsilon \quad \text{and} \quad \langle \boldsymbol{y} - \boldsymbol{y}^*, \nabla_{\boldsymbol{y}} f(\boldsymbol{x}^*, \boldsymbol{y}^*) \rangle \leq \epsilon \quad \forall (\boldsymbol{x}, \boldsymbol{y}) \in \mathcal{X} \times \mathcal{Y}.$$

Definition 1.1 can be equivalently recast as a variational inequality (VI) problem: if $\boldsymbol{z} := (\boldsymbol{x}, \boldsymbol{y})$ and $F : \boldsymbol{z} \mapsto F(\boldsymbol{z}) := (\nabla_{\boldsymbol{x}} f(\boldsymbol{x}, \boldsymbol{y}), -\nabla_{\boldsymbol{y}} f(\boldsymbol{x}, \boldsymbol{y}))$, we are searching for a point $\boldsymbol{z}^* \in \mathcal{Z} := \mathcal{X} \times \mathcal{Y}$ such that $\langle \boldsymbol{z} - \boldsymbol{z}^*, F(\boldsymbol{z}^*) \rangle \geq -2\epsilon$ for all $\boldsymbol{z} \in \mathcal{Z}$. Yet another equivalent definition is instead based on approximate *fixed points* of gradient descent/ascent (GDA); namely, Definition 1.1 amounts to bounding the gradient mappings

$$\|\boldsymbol{x}^* - \Pi_{\mathcal{X}}(\boldsymbol{x}^* - \nabla_{\boldsymbol{x}} f(\boldsymbol{x}^*, \boldsymbol{y}^*))\| \leq \epsilon', \|\boldsymbol{y}^* - \Pi_{\mathcal{Y}}(\boldsymbol{y}^* + \nabla_{\boldsymbol{y}} f(\boldsymbol{x}^*, \boldsymbol{y}^*))\| \leq \epsilon' \quad \text{(Fixed points of GDA)}$$

for some approximation parameter $\epsilon' > 0$ that is (polynomially) dependent on $\epsilon > 0$, where $\|\cdot\|$ is the (Euclidean) $\ell_2$ norm and $\Pi(\cdot)$ is the projection operator. Other definitions that differentiate between the order of play between players—based on the notion of a Stackelberg equilibrium—have also been considered in the literature [Jin et al., 2020].

The complexity of min-max optimization is well-understood in certain special cases, such as when $f$ is convex-concave (*e.g.*, Korpelevich [1976], Mertikopoulos et al. [2019], Cai et al. [2022], Choudhury et al. [2023], Gorbunov et al. [2022], and references therein), or more broadly, nonconvex-concave [Lin et al., 2020, Xu et al., 2023, Luo et al., 2020]. However, the complexity of general min-max optimization problems, when the objective function $f$ is nonconvex-nonconcave, has remained wide open despite intense efforts in recent years. Daskalakis et al. [2021] made progress by establishing certain hardness results targeting the more challenging setting in which there is a *joint* (that is, coupled) set of constraints. In fact, it turns out that their lower bounds apply even for linear-nonconcave objective functions (*cf.* Bernasconi et al. [2024]), showing that their hardness result is driven by the presence of joint constraints—indeed, under uncoupled constraints, many efficient algorithms attaining Definition 1.1 (for linear-nonconcave problems) have been documented in the literature. In the context of min-max optimization, the most well-studied setting posits that players have independent constraints; this is the primary focus of our paper.

## 1.1 Our results

We establish new complexity lower bounds in min-max optimization for computing equilibria in the sense of Definition 1.1; our main results are gathered in Table 1.

Table 1: The main results of this paper. NE stands for Nash equilibrium and FONE for first-order Nash equilibrium (Definition 1.1). We also abbreviate symmetric to "sym." (second column).

| Class of problems | Eq. concept | Complexity | Even for |
|---|---|---|---|
| Adversarial team games | $\epsilon$-NE | CLS-complete (Theorem 1.3) | 3-player (2 vs 1), polymatrix |
| Symmetric min-max | sym. $\epsilon$-FONE | PPAD-complete (Theorem 1.7) | polymatrix, team 0-sum, $\epsilon = 1/n^c$ |
| | non-sym. $\epsilon$-FONE | NP-hard (Theorem 1.6) | quadratics, $\epsilon = 1/n^c$ |

**Adversarial team games**  We first examine an important special case of (1): *adversarial team games* [von Stengel and Koller, 1997]. Here, a team of $n$ players with identical interests is competing against a single adversarial player. (In such settings, Definition 1.1 captures precisely the Nash equilibria of the game.) The computational complexity of this problem was placed by Anagnostides et al. [2023] in the complexity class CLS—which stands for continuous local search [Daskalakis and Papadimitriou, 2011]. Further, by virtue of a result of Babichenko and Rubinstein [2021], computing Nash equilibria in adversarial team games when $n \gg 1$ is CLS-complete. In the context of this prior work, an important question left open by Anagnostides et al. [2023] concerns the case where $n$ is a small constant, a regime not captured by the hardness result of Babichenko and Rubinstein [2021]

pertaining to identical-interest games—in such games, one can simply identify the strategy leading to the highest payoff, which is tractable when $n$ is small.

We show that even when $n = 2$, computing an $\epsilon$-Nash equilibrium in adversarial team games is CLS-complete. (The case where $n = 1$ amounts to two-player zero-sum games, known to be in P.)

**Theorem 1.2.** *Computing an $\epsilon$-Nash equilibrium in $3$-player (that is, $2$ vs. $1$) adversarial team games is* CLS-*complete.*

Coupled with earlier results, Theorem 1.2 completely characterizes the complexity landscape for computing Nash equilibria in adversarial team games.

Our proof is based on a recent hardness result of Ghosh and Hollender [2024] (*cf.* Tewolde et al. [2025]), who proved that computing a *symmetric* $\epsilon$-Nash equilibrium in a symmetric two-player game with identical payoffs is CLS-complete. The key idea in our reduction is that one can leverage the adversarial player so as to enforce symmetry between the team players, without affecting the equilibria of the original game; the basic gadget underpinning this reduction is analyzed in Section 3.1.

Incidentally, our CLS-hardness reduction hinges on a *polymatrix* adversarial team game, thereby addressing another open question left recently by Hollender et al. [2025].

**Theorem 1.3.** *Theorem 1.2 holds even when one restricts to polymatrix, $3$-player adversarial team games.*

We complement the above hardness result by further characterizing the complexity of deciding whether an adversarial team game admits a unique (approximate) Nash equilibrium (Theorem 3.5).

**Symmetric min-max optimization**   As we have seen, symmetry plays a key role in the proof of Theorem 1.2, but that result places no restrictions on whether the equilibrium is symmetric or not—this is indeed the crux of the argument. The next problem we consider concerns computing *symmetric* equilibria in *symmetric* min-max optimization problems, in the following natural sense.

**Definition 1.4** (Symmetric min-max optimization)**.** A function $f : \mathcal{X} \times \mathcal{Y} \to \mathbb{R}$ is called *antisymmetric* if $\mathcal{X} = \mathcal{Y}$ and

$$f(\boldsymbol{x}, \boldsymbol{y}) = -f(\boldsymbol{y}, \boldsymbol{x}) \quad \forall (\boldsymbol{x}, \boldsymbol{y}) \in \mathcal{X} \times \mathcal{Y}.$$

Furthermore, a point $(\boldsymbol{x}, \boldsymbol{y}) \in \mathcal{X} \times \mathcal{Y}$ is called *symmetric* if $\boldsymbol{x} = \boldsymbol{y}$. The associated min-max optimization problem is called symmetric if the underlying function $f$ is *antisymmetric*.[2]

Symmetric zero-sum games are ubiquitous in the literature and in practical applications alike. Many popular recreational games used for AI benchmarking, such as poker and battleship, are symmetric when roles are assigned at random; the symmetry assumption is particularly natural as it ensures that no player has an *a priori* advantage before the game begins.

The study of symmetric equilibria has a long history in the development of game theory, propelled by Nash's pathbreaking PhD thesis [Nash, 1950] (*cf.* Gale et al. [1951]), and has remained a popular research topic ever since [Tewolde et al., 2025, Emmons et al., 2022, Garg et al., 2018, Ghosh and Hollender, 2024, Mehta, 2014]. Classic examples in game theory, including rock–paper–scissors and matching pennies, are also symmetric; these games were already discussed in the original work of von Neumann [1928].

It is not hard to see that symmetric min-max optimization problems, in the sense of Definition 1.4, always admit *symmetric* first-order Nash equilibria. What is more, we show that computing such a symmetric equilibrium is in the complexity class PPAD [Papadimitriou, 1994]; this is based on an argument of Etessami and Yannakakis [2010], and complements Daskalakis et al. [2021], who proved that the problem of computing approximate fixed points of gradient descent/ascent—which they refer to as GDAFIXEDPOINT—lies in PPAD. In a celebrated series of work, it was shown that PPAD captures the complexity of computing Nash equilibria in finite games [Daskalakis et al., 2009, Chen et al., 2009]. In this context, we establish that PPAD also characterizes the complexity of computing symmetric first-order Nash equilibria in symmetric min-max optimization problems:

**Theorem 1.5.** *Computing a symmetric $1/n^c$-approximate first-order Nash equilibrium in symmetric $n$-dimensional min-max optimization is* PPAD-*complete for any constant $c > 0$.*

---

[2]The nomenclature of this definition is consistent with the usual terminology in the context of (two-player) zero-sum games: a symmetric zero-sum game is one in which the the underlying game matrix $\mathbf{A}$ is antisymmetric (that is, skew-symmetric), so that $\langle \boldsymbol{x}, \mathbf{A}\boldsymbol{y} \rangle = -\langle \boldsymbol{y}, \mathbf{A}\boldsymbol{x} \rangle$ for all $(\boldsymbol{x}, \boldsymbol{y})$.

Barring major complexity breakthroughs, Theorem 1.5 precludes the existence of algorithms with complexity polynomial in the dimension and $1/\epsilon$, where $\epsilon > 0$ measures the precision (per Definition 1.1), under the symmetry constraint of Definition 1.4. This stands in contrast to (nonconvex) minimization problems, wherein gradient descent converges to stationary points at a rate of $\text{poly}(1/\epsilon)$; even in the regime where $\epsilon = 1/\exp(n)$, computing a stationary point of a smooth function is in CLS [Daskalakis and Papadimitriou, 2011], which is a subclass of PPAD [Fearnley et al., 2023]. In fact, our reduction also rules out the existence of polynomial-time algorithms even when $\epsilon = \Theta(1)$ under some well-believed complexity assumptions (Corollary 4.3).

The proof of Theorem 1.5 is elementary, and is based on the PPAD-hardness of computing *symmetric* Nash equilibria in *symmetric* two-player games. Importantly, our reduction gives an immediate, and significantly simpler, proof (Theorem 4.4) of the PPAD-hardness result of Daskalakis et al. [2021], while being applicable even with respect to quadratic and anti-symmetric functions defined on a product of simplexes.

**Independent and concurrent work**   Bernasconi et al. [2024] also considerably simplified the proof of Daskalakis et al. [2021]. Our hardness result hinges on the intermediate problem of finding a Nash equilibrium in symmetric two-player games [Chen et al., 2009], whereas Bernasconi et al. [2024] showed their hardness result via the problem of finding a Nash equilibrium in (multi-player) polymatrix two-action games. The main qualitative difference between the two is that ours applies to simplex domains while the result of Bernasconi et al. [2024] to box domains. The basic idea of both reductions then is that one can enforce the symmetry constraint $\boldsymbol{x} \approx \boldsymbol{y}$ via coupled constraints.

As a byproduct of Theorem 1.5 and the result of Bernasconi et al. [2024], it follows that any *symmetric* dynamics—whereby both players follow the same online algorithm, as formalized in Definition 4.5—cannot converge to a first-order Nash equilibrium in polynomial time, subject to PPAD $\neq$ P (Theorem 4.6). This already captures many natural dynamics for which prior papers in the literature (*e.g.*, Kalogiannis et al. [2023b]) have painstakingly shown lack of convergence; Theorem 4.6 provides a complexity-theoretic justification for such prior results, while precluding a much broader family of algorithms.

**The complexity of non-symmetric equilibria**   Remaining on symmetric min-max optimization, one natural question arising from Theorem 1.5 concerns the complexity of *non-symmetric* equilibria— defined as having distance at least $\delta > 0$. Unlike their symmetric counterparts, non-symmetric first-order Nash equilibria are not guaranteed to exist. In fact, we establish the following result.

**Theorem 1.6.** *For a symmetric min-max optimization problem, constants $c_1, c_2 > 0$, and $\epsilon = n^{-c_1}$, it is* NP-*hard to distinguish between the following two cases under the promise that one of them holds:*

- *any $\epsilon$-first-order Nash equilibrium $(\boldsymbol{x}^*, \boldsymbol{y}^*)$ satisfies $\|\boldsymbol{x}^* - \boldsymbol{y}^*\| \leq n^{-c_2}$, and*
- *there is an $\epsilon$-first-order Nash equilibrium $(\boldsymbol{x}^*, \boldsymbol{y}^*)$ such that $\|\boldsymbol{x}^* - \boldsymbol{y}^*\| \geq \Omega(1)$.*

The main technical piece is Theorem 4.7, which concerns symmetric, identical-interest, two-player games. It significantly refines the hardness result of McLennan and Tourky [2010] by accounting even for $\text{poly}(1/n)$-Nash equilibria.

**Team zero-sum games**   Finally, building on the reduction of Theorem 1.5 coupled with the gadget behind Theorem 1.2, we establish similar complexity results for team zero-sum games, which generalize adversarial team games by allowing the presence of multiple adversaries. In particular, a *symmetric* two-team zero-sum game and a *symmetric* equilibrium thereof are in accordance with Definition 1.4—no symmetry constraints are imposed within the same team, but only across teams. We obtain a result significantly refining Theorem 1.5.

**Theorem 1.7.** *Computing a symmetric $1/n^c$-Nash equilibrium in symmetric, 6-player (3 vs. 3) team zero-sum polymatrix games is* PPAD-*complete for some constant $c > 0$.*

Unlike our reduction in Theorem 1.5 that comprises quadratic terms, the crux in team zero-sum games is that one needs to employ solely multilinear terms. The basic idea is to again use the gadget underpinning Theorem 1.2, which enforces symmetry without affecting the equilibria of the game, thereby (approximately) reproducing the objective function that establishes Theorem 1.5.

It is interesting to note that the class of polymatrix games we construct to prove Theorem 1.7 belongs to a certain family introduced by Cai and Daskalakis [2011]: one can partition the players into 2

groups so that any pairwise interaction between players of the same group is a coordination game, whereas any pairwise interaction across groups is a zero-sum game. Cai and Daskalakis [2011] showed that computing a Nash equilibrium is PPAD-hard in the more general case where there are 3 groups of players. While the complexity of that problem under 2 groups remains wide open, Theorem 1.7 shows PPAD-hardness for computing *symmetric* Nash equilibria in such games.

Taken together, our results bring us closer to characterizing the complexity of computing equilibria in min-max optimization.

## 1.2 Further related work

Adversarial team games have been the subject of much research tracing back to the influential work of von Stengel and Koller [1997], who introduced the concept of a *team maxmin equilibrium (TME)*; a TME can be viewed as the best Nash equilibrium for the team. Notwithstanding its intrinsic appeal, it turns out that computing a TME is FNP-hard [Borgs et al., 2010]. Indeed, unlike two-player zero-sum games, team zero-sum games generally exhibit a *duality gap*—characterized in the work of Schulman and Vazirani [2017].

This realization has shifted the focus of contemporary research to exploring more permissive solution concepts. One popular such relaxation is *TMECor*, which enables team players to *ex ante* correlate their strategies [Zhang et al., 2023, 2021, Basilico et al., 2017, Carminati et al., 2022, Farina et al., 2018, Zhang and An, 2020, Celli and Gatti, 2018]. Yet, in the context of extensive-form games, computing a TMECor remains intractable; Zhang et al. [2023] provided an exact characterization of its complexity. Team zero-sum games can be thought of as two-player zero-sum games but with *imperfect recall*, and many natural problems immediately become hard without perfect recall (*e.g.*, Tewolde et al. [2023]). Parameterized algorithms have been developed for computing a TMECor based on some natural measure of shared information [Zhang et al., 2023, Carminati et al., 2022]. Beyond adversarial team games, Carminati et al. [2023] recently explored *hidden-role* games, wherein there is uncertainty regarding which players belong in the same team, a feature that often manifests itself in popular recreational games—and used certain cryptographic primitives to solve them.

In contrast, this paper focuses on the usual Nash equilibrium concept, being thereby orthogonal to the above line of work. One drawback of Nash equilibria in adversarial team games is that the (worst-case) value of the team can be significantly lower compared to TME [Basilico et al., 2017]. On the other hand, Anagnostides et al. [2023] showed that $\epsilon$-Nash equilibria in adversarial team games admit an FPTAS, which stands in stark contrast to TME, and indeed, Nash equilibria in general games [Daskalakis et al., 2009, Chen et al., 2009]. This was further strengthened by Kalogiannis et al. [2023a, 2024] for computing $\epsilon$-Nash equilibria in adversarial team *Markov* games—the natural generalization to Markov (aka. stochastic) games.

Related to Definition 1.1 is the natural notion of a *local* min-max equilibrium [Daskalakis and Panageas, 2018, Daskalakis et al., 2021]. It is easy to see that any local min-max equilibrium—with respect to a sufficiently large neighborhood of $(\boldsymbol{x}^*, \boldsymbol{y}^*)$—must satisfy Definition 1.1 [Daskalakis et al., 2021]. Unlike first-order Nash equilibria, local min-max equilibria are not guaranteed to exist.

Finally, Mehta et al. [2015] showed that in two-player symmetric games, deciding whether a non-symmetric Nash equilibrium exists is NP-hard, which directly relates to our Theorem 1.6.

## 2 Preliminaries

**Notation** We use boldface lowercase letters, such as $\boldsymbol{x}, \boldsymbol{y}, \boldsymbol{z}$, to represent vectors, and boldface capital letters, such as $\mathbf{A}, \mathbf{C}$, for matrices. We denote by $x_i$ the $i$th coordinate of a vector $\boldsymbol{x} \in \mathbb{R}^n$. We use the shorthand notation $[n] := \{1, 2, \ldots, n\}$. $\Delta^n := \{\boldsymbol{x} \in \mathbb{R}^n_{\geq 0} : \sum_{i=1}^n x_i = 1\}$ is the probability simplex on $\mathbb{R}^n$. For $i \in [n]$, $\boldsymbol{e}_i \in \Delta^n$ is the $i$th unit vector. $\langle \cdot, \cdot \rangle$ denotes the inner product. For a vector $\boldsymbol{x} \in \mathbb{R}^n$, $\|\boldsymbol{x}\|_2 = \sqrt{\langle \boldsymbol{x}, \boldsymbol{x} \rangle}$ is its Euclidean norm. For $m \leq n$, $\boldsymbol{x}_{[1 \cdots m]} \in \mathbb{R}^m$ is the vector containing the first $m$ coordinates of $\boldsymbol{x}$. We sometimes use the $O(\cdot), \Theta(\cdot), \Omega(\cdot)$ notation to suppress absolute constants. A continuously differentiable function $f$ is *L-smooth* if its gradient is $L$-Lipschitz continuous with respect to $\| \cdot \|_2$; that is, $\|\nabla f(\boldsymbol{x}) - \nabla f(\boldsymbol{x}')\|_2 \leq L \|\boldsymbol{x} - \boldsymbol{x}'\|_2$ for all $\boldsymbol{x}, \boldsymbol{x}'$.

**Two-player games** In a two-player game, represented in normal-form game, each player has a finite set, let $[n]$, of actions. Under a pair of actions $(i, j) \in [n] \times [n]$, the utility of the *row* player is given by $\mathbf{R}_{i,j}$, where $\mathbf{R} \in \mathbb{Q}^{n \times n}$ is the payoff matrix of the row player. Further, we let $\mathbf{C} \in \mathbb{Q}^{n \times n}$ be the payoff matrix of the column player. Players are allowed to randomize by selecting mixed strategies—points in $\Delta^n$. Under a pair of mixed strategies $(\boldsymbol{x}, \boldsymbol{y}) \in \Delta^n \times \Delta^n$, the expected utility of the players is given by $\langle \boldsymbol{x}, \mathbf{R}\boldsymbol{y} \rangle$ and $\langle \boldsymbol{x}, \mathbf{C}\boldsymbol{y} \rangle$, respectively. The canonical solution concept in such games is the *Nash equilibrium* [Nash, 1951], which is recalled below.

**Definition 2.1.** A pair of strategies $(\boldsymbol{x}^*, \boldsymbol{y}^*)$ is an $\epsilon$-*Nash equilibrium* of $(\mathbf{R}, \mathbf{C})$ if

$$\langle \boldsymbol{x}^*, \mathbf{R}\boldsymbol{y}^* \rangle \geq \langle \boldsymbol{x}, \mathbf{R}\boldsymbol{y}^* \rangle - \epsilon \quad \text{and} \quad \langle \boldsymbol{x}^*, \mathbf{C}\boldsymbol{y}^* \rangle \geq \langle \boldsymbol{x}^*, \mathbf{C}\boldsymbol{y} \rangle - \epsilon \quad \forall (\boldsymbol{x}, \boldsymbol{y}) \in \Delta^n \times \Delta^n.$$

**Symmetric two-player games** One of our reductions is based on *symmetric* two-player games, meaning that $\mathbf{R} = \mathbf{C}^\top$. A basic fact is that any symmetric game admits a *symmetric* Nash equilibrium $(\boldsymbol{x}^*, \boldsymbol{x}^*)$. Further, computing a Nash equilibrium in a general game can be reduced to computing a symmetric Nash equilibrium in a symmetric game [Nisan et al., 2007, Theorem 2.4]. In conjunction with the hardness result of Chen et al. [2009], we state the following consequence.

**Theorem 2.2** (Chen et al., 2009)**.** *Computing a symmetric $1/n^c$-Nash equilibrium in a symmetric two-player game is* PPAD*-hard for any constant $c > 0$.*

**Team zero-sum games** A (two-)team zero-sum game is a multi-player game—represented in normal form for the purposes of this paper—in which the players' utilities have a certain structure; namely, we can partition the players into two (disjoint) subsets, such that each player within the same team shares the same utility, whereas players in different teams have opposite utilities—under any possible combination of actions. An *adversarial* team game is a specific type of team zero-sum game wherein one team consists of a single player. As in Definition 2.1 for two-player games, an $\epsilon$-*Nash equilibrium* is a tuple of strategies such that no unilateral deviation yields more than an $\epsilon$ additive improvement in the utility of the deviator.

# 3 Complexity of adversarial team games

We begin by examining equilibrium computation in adversarial team games.

## 3.1 CLS-completeness for 3-player games

Computing $\epsilon$-Nash equilibria in adversarial team games was placed in CLS by Anagnostides et al. [2023], but whether CLS tightly characterizes the complexity of that problem remained open—that was only known when the number of players is large, so that the hardness result of Babichenko and Rubinstein [2021] can kick in. Our reduction here answers this question in the affirmative.

We rely on a recent hardness result of Ghosh and Hollender [2024] concerning symmetric, two-player games with identical payoffs. We summarize their main result below.

**Theorem 3.1** (Ghosh and Hollender, 2024)**.** *Computing an $\epsilon$-Nash equilibrium in a symmetric, identical-payoffs, two-player game is* CLS*-complete.*

Now, let $\mathbf{A} \in \mathbb{Q}^{n \times n}$ be the common payoff matrix of a two-player game, which satisfies $\mathbf{A} = \mathbf{A}^\top$ so that the game is symmetric. Without loss of generality, we will assume that $\mathbf{A}_{i,j} \leq -1$ for all $i, j \in [n]$. We denote by $\mathbf{A}_{\min}$ and $\mathbf{A}_{\max}$ the minimum and maximum entry of $\mathbf{A}$, respectively (which satisfy $\mathbf{A}_{\max}, \mathbf{A}_{\min} \leq -1$). The basic idea of our proof is to suitably use the adversarial player so as to force the other two players to play roughly the same strategy (Lemma 3.2), while (approximately) maintaining the structure of the game (Lemma 3.3). The formal proofs are in Section A.1.

**Definition of the adversarial team game** Based on $\mathbf{A}$, we construct a 3-player adversarial team game as follows. The utility function of the adversary reads

$$u(\boldsymbol{x}, \boldsymbol{y}, \boldsymbol{z}) := \langle \boldsymbol{x}, \mathbf{A}\boldsymbol{y} \rangle + \frac{|\mathbf{A}_{\min}|}{\epsilon} \sum_{i=1}^{n} (z_i(x_i - y_i) + z_{n+i}(y_i - x_i)) + z_{2n+1}|\mathbf{A}_{\min}|. \tag{2}$$

The adversary selects a strategy $\boldsymbol{z} \in \Delta^{2n+1}$, while the team players, who endeavor to minimize (2), select strategies $\boldsymbol{x} \in \Delta^n$ and $\boldsymbol{y} \in \Delta^n$, respectively. (While the range of the utilities in (2) grows with $1/\epsilon$, normalizing to $[-1, 1]$ maintains all of the consequences by suitably adjusting the approximation.)

The first important lemma establishes that, in equilibrium, $\boldsymbol{x} \approx \boldsymbol{y}$. The basic argument proceeds as follows. By construction of (2), the adversary would be able to secure a large payoff whenever there is a coordinate $i \in [n]$ such that $|x_i - y_i| \gg 0$—by virtue of the second term in (2). But that cannot happen in equilibrium, for Player $\boldsymbol{x}$ (or symmetrically Player $\boldsymbol{y}$) can simply neutralize that term in the adversary's utility by playing $\boldsymbol{x} = \boldsymbol{y}$.

**Lemma 3.2** (Equilibrium forces symmetry). *Consider an $\epsilon^2$-Nash equilibrium $(\boldsymbol{x}^*, \boldsymbol{y}^*, \boldsymbol{z}^*)$ of the adversarial team game (2) with $\epsilon^2 \leq 1/2$. Then, $\|\boldsymbol{x}^* - \boldsymbol{y}^*\|_\infty \leq 2\epsilon$.*

Having established that $\boldsymbol{x} \approx \boldsymbol{y}$, the next step is to make sure that the adversarial player does not distort the original game by much. In particular, we need to make sure that the effect of the second term in (2) is negligible. We do so by showing that $z_{2n+1} \approx 1$ (Lemma 3.3).

The argument here is more subtle; roughly speaking, it goes as follows. Suppose that $z_i \gg 0$ or $z_{n+i} \gg 0$ for some $i \in [n]$. Since Player $\boldsymbol{z}$ is approximately best responding, it would then follow that $|y_i^* - x_i^*| \gg 0$—otherwise Player $\boldsymbol{z}$ would prefer to switch to action $2n+1$. But, if $|y_i^* - x_i^*| \gg 0$, Player $\boldsymbol{x}$ could profitably deviate by reallocating probability mass by either removing from or adding to $i$ (depending on whether $y_i^* - x_i^* > 0$), which leads to a contradiction.

**Lemma 3.3** (Most probability mass in $a_{2n+1}$). *Given any $\epsilon^2$-Nash equilibrium $(\boldsymbol{x}^*, \boldsymbol{y}^*, \boldsymbol{z}^*)$ of the adversarial team game (2) with $\epsilon \leq 1/10$, $z_j \leq 9\epsilon$ for all $j \in [2n]$. In particular, $z_{2n+1} \geq 1 - 18n\epsilon$.*

By combining Lemmas 3.2 and 3.3, we can complete the reduction from symmetric two-player games with common payoffs to 3-player adversarial team games, as stated below.

**Theorem 3.4.** *Given any $\epsilon^2$-Nash equilibrium $(\boldsymbol{x}^*, \boldsymbol{y}^*, \boldsymbol{z}^*)$ in the adversarial team game (2), with $\epsilon \leq 1/10$, $(\boldsymbol{y}^*, \boldsymbol{y}^*)$ is a symmetric $(21n+1)|\mathbf{A}_{\min}|\epsilon$-Nash equilibrium of the symmetric, two-player game $(\mathbf{A}, \mathbf{A})$ (that is, $\mathbf{A} = \mathbf{A}^\top$).*

## 3.2 The complexity of determining uniqueness

Another natural question concerns the complexity of determining whether an adversarial team game admits a unique Nash equilibrium. Our next theorem establishes NP-hardness for a version of that problem that accounts for approximate Nash equilibria.

**Theorem 3.5.** *For polymatrix, 3-player adversarial team games, constants $c_1, c_2 > 0$, and $\epsilon = n^{-c_1}$, it is NP-hard to distinguish between the following two cases under the promise that one of them holds:*

- *any two $\epsilon$-Nash equilibria have $\ell_1$-distance at most $n^{-c_2}$, and*
- *there are two $\epsilon$-Nash equilibria that have $\ell_1$-distance $\Omega(1)$.*

We will discuss more about the proof of this theorem later in Section 4.2 when we examine the complexity of computing non-symmetric equilibria in symmetric min-max optimization problems. It is also interesting to point out that an adversatial team game can have a unique Nash equilibrium supported on *irrational* numbers, as we show in Section A.2.

# 4 Complexity of equilibria in symmetric min-max optimization

This section characterizes the complexity of computing symmetric first-order Nash equilibria (Definition 1.1) in symmetric min-max optimization problems in the sense of Definition 1.4; namely, when $f(\boldsymbol{x}, \boldsymbol{y}) = -f(\boldsymbol{y}, \boldsymbol{x})$ for all $(\boldsymbol{x}, \boldsymbol{y}) \in \mathcal{X} \times \mathcal{Y}$ and $\mathcal{X} = \mathcal{Y}$.

## 4.1 Problem definitions and hardness results for symmetric equilibria

Given a continuously differentiable function $f : \mathcal{D} \to \mathbb{R}$, we set $F_{\mathrm{GDA}} : \mathcal{D} \to \mathcal{D}$ to be

$$F_{\mathrm{GDA}}(\boldsymbol{x}, \boldsymbol{y}) := \prod_{\mathcal{D}} [\boldsymbol{x} - \nabla_{\boldsymbol{x}} f(\boldsymbol{x}, \boldsymbol{y}), \boldsymbol{y} + \nabla_{\boldsymbol{y}} f(\boldsymbol{x}, \boldsymbol{y})] \text{ for } (\boldsymbol{x}, \boldsymbol{y}) \in \mathcal{D},$$

the norm of which measures the fixed-point gap and corresponds to the update rule of GDA with stepsize equal to one; we recall that Player $\boldsymbol{x}$ is the minimizer, while Player $\boldsymbol{y}$ is the maximizer. The domain $\mathcal{D}$ is a compact subset of $\mathbb{R}^d$ for some $d \in \mathbb{N}$. Moreover, the projection operator $\prod$ is applied

jointly on $\mathcal{D}$.[3] When $\mathcal{D}$ can be expressed as a *Cartesian* product $\mathcal{X} \times \mathcal{Y}$, the domain set is called *uncoupled* (and the projection can be done independently), otherwise it is called *coupled* (or *joint*).

We begin by introducing the problem of computing fixed points of gradient descent/ascent (GDA) for domains expressed as the Cartesian product of polytopes, modifying the computational problem GDAFIXEDPOINT introduced by Daskalakis et al. [2021].

**GDAFIXEDPOINT Problem.**
INPUT:

- Precision parameter $\epsilon > 0$ and smoothness parameter $L$,
- Polynomial-time Turing machine $\mathcal{C}_f$ evaluating a $L$-smooth function $f : \mathcal{X} \times \mathcal{Y} \to \mathbb{R}$ and its gradient $\nabla f : \mathcal{X} \times \mathcal{Y} \to \mathbb{R}^d$, where $\mathcal{X} = \{\boldsymbol{x} : \mathbf{A}_x \boldsymbol{x} \leq \boldsymbol{b}_x\}$ and $\mathcal{Y} = \{\boldsymbol{y} : \mathbf{A}_y \boldsymbol{y} \leq \boldsymbol{b}_y\}$ are nonempty, bounded polytopes described by input matrices $\mathbf{A}_x \in \mathbb{R}^{m_x \times d_x}$, $\mathbf{A}_y \in \mathbb{R}^{m_y \times d_y}$ and vectors $\boldsymbol{b}_x \in \mathbb{R}^{m_x}$, $\boldsymbol{b}_y \in \mathbb{R}^{m_y}$, with $d := d_x + d_y$.

OUTPUT: A point $(\boldsymbol{x}^*, \boldsymbol{y}^*) \in \mathcal{X} \times \mathcal{Y}$ such that $\|(\boldsymbol{x}^*, \boldsymbol{y}^*) - F_{GDA}(\boldsymbol{x}^*, \boldsymbol{y}^*)\|_2 \leq \epsilon$.

Based on GDAFIXEDPOINT, we introduce the problem SYMGDAFIXEDPOINT, which captures the problem of computing *symmetric* (approximate) fixed points of GDA for symmetric min-max optimization problems. We define our computational problems as promise problems.

**SYMGDAFIXEDPOINT Problem.**
INPUT:

- Precision parameter $\epsilon > 0$ and smoothness parameter $L$,
- Polynomial-time Turing machine $\mathcal{C}_f$ evaluating a $L$-smooth, antisymmetric function $f : \mathcal{X} \times \mathcal{X} \to \mathbb{R}$ and its gradient $\nabla f : \mathcal{X} \times \mathcal{X} \to \mathbb{R}^{2d}$, where $\mathcal{X} = \{\boldsymbol{x} : \mathbf{A}\boldsymbol{x} \leq \boldsymbol{b}\}$ is a nonempty, bounded polytope described by an input matrix $\mathbf{A} \in \mathbb{R}^{m \times d}$ and vector $\boldsymbol{b} \in \mathbb{R}^m$.

OUTPUT: A point $(\boldsymbol{x}^*, \boldsymbol{x}^*) \in \mathcal{X} \times \mathcal{X}$ such that $\|(\boldsymbol{x}^*, \boldsymbol{x}^*) - F_{GDA}(\boldsymbol{x}^*, \boldsymbol{x}^*)\|_2 \leq \epsilon$.

We start by showing that SYMGDAFIXEDPOINT also lies in PPAD; the fact that GDAFIXEDPOINT is in PPAD—even under coupled domains—was shown to be the case by Daskalakis et al. [2021]. The detailed proof is included in the appendix.

**Lemma 4.1.** SYMGDAFIXEDPOINT *is a total search problem and lies in* PPAD.

Having established that SYMGDAFIXEDPOINT belongs in PPAD, we now state the first main hardness result of this section.

**Theorem 4.2** (Complexity for symmetric equilibrium). SYMGDAFIXEDPOINT *is* PPAD-*complete, even for quadratic functions.*

The basic idea of the proof is to consider the objective

$$f(\boldsymbol{x}, \boldsymbol{y}) := \frac{1}{2}\langle \boldsymbol{y}, \mathbf{A}\boldsymbol{y} \rangle - \frac{1}{2}\langle \boldsymbol{x}, \mathbf{A}\boldsymbol{x} \rangle + \langle \boldsymbol{y}, \mathbf{C}\boldsymbol{x} \rangle, \tag{3}$$

where $\mathbf{A}$ is symmetric and $\mathbf{C}$ is skew-symmetric. Theorem 4.2 then follows from some elementary calculations, as we show in Section A.3.

For symmetric first-order Nash equilibria, our argument establishes PPAD-hardness for any $\epsilon \leq 1/n^c$, where $c > 0$ (as claimed in Theorem 1.5). Moreover, leveraging the hardness result of Rubinstein [2016], we can also immediately obtain constant inapproximability under the so-called *exponential-time hypothesis (ETH)* for PPAD—which postulates than any algorithm for solving ENDOFALINE, the prototypical PPAD-complete problem, requires $2^{\tilde{\Omega}(n)}$ time.

**Corollary 4.3.** *Computing an* $\Theta(1)$-*approximate first-order Nash equilibrium in symmetric $n$-dimensional min-max optimization requires* $n^{\tilde{\Omega}(\log n)}$ *time, assuming ETH for* PPAD.

---

[3]This is the "safe" version of GDA because it ensures that the mapping always lies in $\mathcal{D}$. One could also project independently on $\mathcal{D}(\boldsymbol{y}) = \{\boldsymbol{x}' : (\boldsymbol{x}', \boldsymbol{y}) \in \mathcal{D}\}$ and $\mathcal{D}(\boldsymbol{x}) = \{\boldsymbol{y}' : (\boldsymbol{x}, \boldsymbol{y}') \in \mathcal{D}\}$; see Daskalakis et al. [2021] for further details and the polynomial equivalence for finding fixed points for both versions.

The argument of Theorem 4.2 can be slightly modified to imply the main result of Daskalakis et al. [2021]—with simplex instead of box constraints—as stated below.

**Theorem 4.4** (PPAD-hardness for coupled domains). *The problem* GDAFIXEDPOINT *is* PPAD-*hard when the domain is a joint polytope, even for quadratic functions.*

The main idea is to add coupled constraints in order to *force symmetry*: $-\delta \leq x_i - y_i \leq \delta$ for all $i \in [n]$, where, if $\epsilon$ is the approximation accuracy, $\delta$ is of order $\Theta\left(\epsilon^{1/4}\right)$. Compared to the equilibrium studied in Daskalakis et al. [2021], the symmetric equilibrium considered in our work is stronger in that it accounts for all deviations, not merely ones on the coupled feasibility set. We present the proof of Theorem 4.4 in Section A.3.

**Hardness results for symmetric dynamics**    Another interesting consequence of Theorem 4.2 is that it precludes convergence under a broad class of algorithms in general min-max optimization.

**Definition 4.5** (Symmetric learning algorithms for min-max). Let $T \in \mathbb{N}$. A deterministic, polynomial-time learning algorithm $\mathcal{A}$ proceeds as follows for any time $t \in [T]$. It outputs a strategy as a function of the history $\mathcal{H}^{(t)}$ it has observed so far (where $\mathcal{H}^{(1)} := \emptyset$), and then receives as feedback $\boldsymbol{g}^{(t)}$. It then updates $\mathcal{H}^{(t+1)} := (\mathcal{H}^{(t)}, \boldsymbol{g}^{(t)})$. A *symmetric* learning algorithm in min-max optimization consists of Player $\boldsymbol{x}$ employing algorithm $\mathcal{A}$ with history $\mathcal{H}_x^{(t)} := (\nabla_{\boldsymbol{x}} f(\boldsymbol{x}^{(t)}, \boldsymbol{y}^{(t)}))_{t=1}^T$, and Player $\boldsymbol{y}$ employing the *same* algorithm with history $\mathcal{H}_y^{(t)} := (-\nabla_{\boldsymbol{y}} f(\boldsymbol{x}^{(t)}, \boldsymbol{y}^{(t)}))_{t=1}^T$.

Note that a consequence of the above definition is that both players initialize from the same strategy. Many natural and well-studied algorithms in min-max optimization adhere to Definition 4.5. Besides the obvious example of gradient descent/ascent, we mention extragradient descent(/ascent), optimistic gradient descent(/ascent), and optimistic multiplicative weights—all assumed to be executed simultaneously. A simple non-example is *alternating* gradient descent(/ascent) [Wibisono et al., 2022, Bailey et al., 2020], wherein players do not update their strategies simultaneously.

**Theorem 4.6.** *No symmetric learning algorithm (per Definition 4.5) can converge to $\epsilon$-first-order Nash equilibria in min-max optimization in polynomial time when $\epsilon = 1/n^c$, unless* PPAD = P.

This is a consequence of our argument in Theorem 4.2: under Definition 4.5 and the min-max optimization problem (3), it follows inductively that $\boldsymbol{x}^{(t)} = \boldsymbol{y}^{(t)}$ and $\mathcal{H}_x^{(t)} = \mathcal{H}_y^{(t)}$ for all $t \in [T]$. But computing a symmetric first-order Nash equilibrium is PPAD-hard when $\epsilon = 1/n^c$ (Theorem 4.2).

Assuming that P $\neq$ PPAD, Theorem 4.6, and in particular its instantiation in team zero-sum games (Theorem 1.7), significantly generalizes some impossibility results shown by Kalogiannis et al. [2023b] concerning certain algorithms, such as optimistic gradient descent(/ascent)—our hardness result goes much further, precluding any algorithm subject to Definition 4.5, albeit being conditional.

## 4.2   The complexity of non-symmetric fixed points

An immediate question raised by Theorem 4.2 concerns the computational complexity of finding *non-symmetric* fixed points of GDA for symmetric min-max optimization problems. Since totality is not guaranteed, unlike SYMGDAFIXEDPOINT, we cannot hope to prove membership in PPAD. In fact, we show that finding a non-symmetric fixed point of GDA is FNP-hard. To do so, we first define formally the computational problem of interest.

> **NONSYMGDAFIXEDPOINT Problem.**
> INPUT:
>
> - Parameters $\epsilon, \delta > 0$ and Lipschitz constant $L$ and
> - Polynomial-time Turing machine $\mathcal{C}_f$ evaluating a $L$-smooth antisymmetric function $f : \mathcal{X} \times \mathcal{X} \to \mathbb{R}$ and its gradient $\nabla f : \mathcal{X} \times \mathcal{X} \to \mathbb{R}^{2d}$, where $\mathcal{X} = \{\boldsymbol{x} : \mathbf{A}\boldsymbol{x} \leq \boldsymbol{b}\}$ is a nonempty, bounded polytope described by a matrix $\mathbf{A} \in \mathbb{R}^{m \times d}$ and vector $\boldsymbol{b} \in \mathbb{R}^m$.
>
> OUTPUT:    A point $(\boldsymbol{x}^*, \boldsymbol{y}^*) \in \mathcal{X} \times \mathcal{X}$ such that $\|\boldsymbol{x}^* - \boldsymbol{y}^*\|_2 \geq \delta$ and $\|(\boldsymbol{x}^*, \boldsymbol{y}^*) - F_{GDA}(\boldsymbol{x}^*, \boldsymbol{y}^*)\|_2 \leq \epsilon$ if it exists, otherwise return NO.

We establish that NONSYMGDAFIXEDPOINT is FNP-hard. Our reduction builds on the hardness result of McLennan and Tourky [2010]—in turn based on earlier work by Gilboa and Zemel [1989],

Conitzer and Sandholm [2008]—which we significantly refine in order to account for $\text{poly}(1/n)$-Nash equilibria. Our result, which forms the basis for Theorem 1.6 and Theorem 3.5, is summarized below.

**Theorem 4.7.** *For symmetric, identical-interest, two-player games, constants $c_1, c_2 > 0$, and $\epsilon = n^{-c_1}$, it is* NP-*hard to distinguish between the following two cases under the promise that one of them holds:*

- *any two symmetric $\epsilon$-Nash equilibria have $\ell_1$-distance at most $n^{-c_2}$, and*
- *there are two symmetric $\epsilon$-Nash equilibria that have $\ell_1$-distance $\Omega(1)$.*

The proof of Theorem 1.6 now follows by considering the antisymmetric function $f(\boldsymbol{x}, \boldsymbol{y}) := \boldsymbol{y}^\top \mathbf{B} \boldsymbol{y} - \boldsymbol{x}^\top \mathbf{B} \boldsymbol{x}$ for a suitable matrix $\mathbf{B}$ (defined per the hard instance from Theorem 4.7 based on $k$-clique). FNP-hardness follows similarly by considering a search version of maximum clique.

Finally, the proof of Theorem 3.5 that was claimed earlier follows immediately by combining Theorem 4.7 with the reduction of Section 3.1, and in particular, Lemmas 3.2 and 3.3.

### 4.3 Team zero-sum games

Our previous hardness result concerning symmetric min-max optimization problems does not have any immediate implications for (normal-form) team zero-sum games since the class of hard instances we constructed earlier contains a quadratic term. Our next result provides such a hardness result by combining the basic gadget we introduced in Section 3.1 in the context of adversarial team games; the basic pieces of the argument are similar to the ones we described in Section 3.1, and so the proof is deferred to Section A.5. Our goal is to prove the following.

**Theorem 1.7.** *Computing a symmetric $1/n^c$-Nash equilibrium in symmetric, 6-player (3 vs. 3) team zero-sum polymatrix games is* PPAD-*complete for some constant $c > 0$.*

Let us describe the class of 3 vs. 3 team zero-sum games upon which our hardness result is based on. Based on (2), we define the auxiliary function

$$\delta : \Delta^n \times \Delta^n \times \Delta^{2n+1} \ni (\boldsymbol{x}, \boldsymbol{y}, \boldsymbol{z}) \mapsto \frac{|\mathbf{A}_{\min}|}{\epsilon} \sum_{i=1}^n (z_i(x_i - y_i) + z_{n+i}(y_i - x_i)) + |\mathbf{A}_{\min}| z_{2n+1}.$$

In what follows, the 3 players of the one team will be identified with $(\boldsymbol{x}, \boldsymbol{y}, \boldsymbol{z})$, while the 3 players of the other team with $(\hat{\boldsymbol{x}}, \hat{\boldsymbol{y}}, \hat{\boldsymbol{z}})$. We define the utility of the latter team to be

$$u(\boldsymbol{x}, \boldsymbol{y}, \boldsymbol{z}, \hat{\boldsymbol{x}}, \hat{\boldsymbol{y}}, \hat{\boldsymbol{z}}) = \langle \boldsymbol{x}, \mathbf{A}\boldsymbol{y} \rangle - \langle \hat{\boldsymbol{x}}, \mathbf{A}\hat{\boldsymbol{y}} \rangle + \langle \boldsymbol{x}, \mathbf{C}\hat{\boldsymbol{x}} \rangle + \delta(\boldsymbol{x}, \boldsymbol{y}, \hat{\boldsymbol{z}}) - \delta(\hat{\boldsymbol{x}}, \hat{\boldsymbol{y}}, \boldsymbol{z}), \qquad (4)$$

where $\mathbf{A}$ is symmetric and $\mathbf{C}$ is skew-symmetric. The rest of the argument follows Section 3.1.

## 5 Conclusion and open problems

We have provided a number of new complexity results concerning min-max optimization in general, and team zero-sum games in particular (see Table 1). There are many interesting avenues for future research. The complexity of computing first-order Nash equilibria (equivalently, the GDAFixedPoint problem) remains wide open, but our hardness results suggest a possible approach: as we have seen, in symmetric min-max optimization, computing *either* symmetric or non-symmetric equilibria is intractable, so it would be enough if one could establish this using the same underlying function—that is, somehow combine our two reductions into one. It would also be interesting to see whether our hardness results can be extended to more structured min-max optimization problems, such as adversarial training and GANs.

## Acknowledgments

I.P. and J.Y. were supported by NSF grant CCF-2454115. I.P. would like to acknowledge ICS research award and a start-up grant from UCI. Part of this work was done while I.P. and J.Y. were visiting Archimedes Research Unit. T.S. is supported by the Vannevar Bush Faculty Fellowship ONR N00014-23-1-2876, National Science Foundation grants RI-2312342 and RI-1901403, ARO award W911NF2210266, and NIH award A240108S001. We are grateful to Alexandros Hollender for many valuable discussions.

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

# A Omitted proofs

This section contains the proofs omitted from the main body.

## A.1 Proofs from Section 3.1

We begin by stating and proving a simple auxiliary lemma.

**Lemma A.1.** *Let $(\boldsymbol{x}_i^*, \boldsymbol{x}_{-i}^*)$ be an $\epsilon^2$-Nash equilibrium of a normal-form game, and $a_j$ any action of player $i$. If $u_i(a_k, \boldsymbol{x}_{-i}^*) \leq u_i(a_j, \boldsymbol{x}_{-i}^*) - c$ for some $c > 0$ and $k \in [n]$, then $x_i^*(a_k) \leq \epsilon^2/c$.*

*Proof.* For the sake of contradiction, suppose that $x_i^*(a_k) > \frac{\epsilon^2}{c}$ for some $k \in [n]$ such that $u_i(a_k, \boldsymbol{x}_{-i}^*) \leq u_i(a_j, \boldsymbol{x}_{-i}^*) - c$. Consider the strategy $\Delta^n \ni \boldsymbol{x}_i' = \boldsymbol{x}_i^* + x_i^*(a_k)\boldsymbol{e}_j - x_i^*(a_k)\boldsymbol{e}_k$. Then, we have

$$
\begin{aligned}
u_i(\boldsymbol{x}_i', \boldsymbol{x}_{-i}^*) - u_i(\boldsymbol{x}_i^*, \boldsymbol{x}_{-i}^*) &= x_i^*(a_k)u_i(a_j, \boldsymbol{x}_{-i}^*) - x_i^*(a_k)u_i(a_k, \boldsymbol{x}_{-i}^*) \\
&\geq cx_i^*(a_k) \\
&> \epsilon^2.
\end{aligned}
$$

That is, deviating to $\boldsymbol{x}_i'$ yields a utility benefit strictly larger than $\epsilon^2$, which contradicts the assumption that $(\boldsymbol{x}_i^*, \boldsymbol{x}_{-i}^*)$ is an $\epsilon^2$-Nash equilibrium. $\qquad\square$

We move on to the proof of Lemma 3.2.

*Proof of Lemma 3.2.* For the sake of contradiction, suppose that $x_i^* - y_i^* > 2\epsilon$ for some $i \in [n]$ (the case where $y_i^* - x_i^* > 2\epsilon$ is symmetric, and can be treated analogously). Player $\boldsymbol{z}$ could then choose action $a_i$ (with probability 1), which secures a utility of

$$
u(\boldsymbol{x}^*, \boldsymbol{y}^*, a_i) = \langle \boldsymbol{x}^*, \mathbf{A}\boldsymbol{y}^* \rangle + \frac{|\mathbf{A}_{\min}|}{\epsilon} \cdot (x_i^* - y_i^*) > \langle \boldsymbol{x}^*, \mathbf{A}\boldsymbol{y}^* \rangle + 2|\mathbf{A}_{\min}|,
$$

since $x_i^* - y_i^* > 2\epsilon$. At the same time, Player $\boldsymbol{z}$ could choose action $a_{2n+1}$, which secures a utility of $u(\boldsymbol{x}^*, \boldsymbol{y}^*, a_{2n+1}) = \langle \boldsymbol{x}^*, \mathbf{A}\boldsymbol{y}^* \rangle + |\mathbf{A}_{\min}|$. So,

$$
u(\boldsymbol{x}^*, \boldsymbol{y}^*, a_i) - u(\boldsymbol{x}^*, \boldsymbol{y}^*, a_{2n+1}) \geq |\mathbf{A}_{\min}|.
$$

Applying Lemma A.1,

$$
z_{2n+1}^* \leq \frac{\epsilon^2}{|\mathbf{A}_{\min}|} \leq \epsilon^2. \tag{5}
$$

Also, using the fact that $(\boldsymbol{x}^*, \boldsymbol{y}^*, \boldsymbol{z}^*)$ is an $\epsilon^2$-Nash equilibrium,

$$
\begin{aligned}
u(\boldsymbol{x}^*, \boldsymbol{y}^*, \boldsymbol{z}^*) &\geq u(\boldsymbol{x}^*, \boldsymbol{y}^*, a_i) - \epsilon^2 \\
&\geq \langle \boldsymbol{x}^*, \mathbf{A}\boldsymbol{y}^* \rangle + 2|\mathbf{A}_{\min}| - \epsilon^2 \\
&\geq \mathbf{A}_{\min} + 2|\mathbf{A}_{\min}| - \epsilon^2 \\
&= |\mathbf{A}_{\min}| - \epsilon^2, \tag{6}
\end{aligned}
$$

since we have assumed that $\mathbf{A}_{\min} < 0$. Now, consider the deviation of Player $\boldsymbol{x}$ (from strategy $\boldsymbol{x}^*$) to $\boldsymbol{x}' := \boldsymbol{y}^*$. Then, $u(\boldsymbol{x}', \boldsymbol{y}^*, \boldsymbol{z}^*) = \langle \boldsymbol{y}^*, \mathbf{A}\boldsymbol{y}^* \rangle + z_{2n+1}^* \cdot |\mathbf{A}_{\min}|$. Thus, combining with (6) and (5),

$$
\begin{aligned}
u(\boldsymbol{x}', \boldsymbol{y}^*, \boldsymbol{z}^*) - u(\boldsymbol{x}^*, \boldsymbol{y}^*, \boldsymbol{z}^*) &\leq \langle \boldsymbol{y}^*, \mathbf{A}\boldsymbol{y}^* \rangle + z_{2n+1}^* \cdot |\mathbf{A}_{\min}| - |\mathbf{A}_{\min}| + \epsilon^2 \\
&\leq \langle \boldsymbol{y}^*, \mathbf{A}\boldsymbol{y}^* \rangle + \epsilon^2 \cdot |\mathbf{A}_{\min}| - |\mathbf{A}_{\min}| + \epsilon^2 \\
&\leq \mathbf{A}_{\max} - |\mathbf{A}_{\min}| + \epsilon^2(|\mathbf{A}_{\min}| + 1) \\
&\leq -1 < -\epsilon^2, \tag{7}
\end{aligned}
$$

where we used that $\mathbf{A}_{\max}, \mathbf{A}_{\min} \leq -1$ and $\epsilon^2 \leq 1/2$. But (7) contradicts the fact that $(\boldsymbol{x}^*, \boldsymbol{y}^*, \boldsymbol{z}^*)$ is an $\epsilon^2$-Nash equilibrium since deviating to $\boldsymbol{x}'$ yields a utility improvement (equivalently, decrease in cost) strictly larger than $\epsilon^2$. This completes the proof. $\qquad\square$

We proceed with the proof of Lemma 3.3.

*Proof of Lemma 3.3.* From Lemma 3.2 it holds that $\|\boldsymbol{x}^* - \boldsymbol{y}^*\|_\infty \leq 2\epsilon$. Let $i \in [n]$. We assume that $i$ is such that $x_i^* - y_i^* \geq 0$; the contrary case is symmetric. We consider two cases. First, suppose that $|x_i^* - y_i^*| \leq \epsilon/2$. Then, we have

$$u(\boldsymbol{x}^*, \boldsymbol{y}^*, a_{2n+1}) - u(\boldsymbol{x}^*, \boldsymbol{y}^*, a_i) \geq |\mathbf{A}_{\min}| - \frac{|\mathbf{A}_{\min}|}{\epsilon}(x_i^* - y_i^*)$$
$$\geq \frac{1}{2}|\mathbf{A}_{\min}| \geq \frac{1}{2}.$$

Thus, by Lemma A.1, we conclude that $z_i \leq 2\epsilon^2$. Similarly,

$$u(\boldsymbol{x}^*, \boldsymbol{y}^*, a_{2n+1}) - u(\boldsymbol{x}^*, \boldsymbol{y}^*, a_{n+i}) \geq |\mathbf{A}_{\min}| - \frac{|\mathbf{A}_{\min}|}{\epsilon}(y_i^* - x_i^*) \geq |\mathbf{A}_{\min}| \geq 1,$$

since $x_i^* - y_i^* \geq 0$. Again, Lemma A.1 implies that $z_{n+i} \leq \epsilon^2$.

It thus suffices to treat the case where $|x_i^* - y_i^*| > \epsilon/2$ (assuming that $x_i^* - y_i^* \geq 0$). It follows that there exists $j \in [n]$ such that $x_j^* - y_j^* < 0$. In addition, we observe that $u(\boldsymbol{x}^*, \boldsymbol{y}^*, a_{2n+1}) = \langle \boldsymbol{x}^*, \mathbf{A}\boldsymbol{y}^* \rangle + |\mathbf{A}_{\min}| \geq \langle \boldsymbol{x}^*, \mathbf{A}\boldsymbol{y}^* \rangle + 1$, whereas $u(\boldsymbol{x}^*, \boldsymbol{y}^*, a_j) < \langle \boldsymbol{x}^*, \mathbf{A}\boldsymbol{y}^* \rangle$ and $u(\boldsymbol{x}^*, \boldsymbol{y}^*, a_{n+i}) < \langle \boldsymbol{x}^*, \mathbf{A}\boldsymbol{y}^* \rangle$. As a result, Lemma A.1 implies that $z_{n+i}^* \leq \epsilon^2$ and $z_j^* \leq \epsilon^2$.

Now, consider the deviation

$$\Delta^n \ni \boldsymbol{x}' = \boldsymbol{x}^* + (y_i^* - x_i^*)\boldsymbol{e}_i + (x_i^* - y_i^*)\boldsymbol{e}_j;$$

that is, $\boldsymbol{x}'$ is the strategy that results from $\boldsymbol{x}$ by reallocating $(x_i^* - y_i^*)$ probability mass from action $a_i$ to action $a_j$. The difference $u(\boldsymbol{x}', \boldsymbol{y}^*, \boldsymbol{z}^*) - u(\boldsymbol{x}^*, \boldsymbol{y}^*, \boldsymbol{z}^*)$ can be expressed as

$$\langle \boldsymbol{x}' - \boldsymbol{x}^*, \mathbf{A}\boldsymbol{y}^* \rangle + \frac{|\mathbf{A}_{\min}|}{\epsilon}\left(z_i^*(x_i' - x_i^*) + z_j^*(x_j' - x_j^*) + z_{n+i}^*(x_i^* - x_i') + z_{n+j}^*(x_j^* - x_j')\right)$$
$$= \langle \boldsymbol{x}' - \boldsymbol{x}^*, \mathbf{A}\boldsymbol{y}^* \rangle + \frac{|\mathbf{A}_{\min}|}{\epsilon}\left(z_i^*(y_i^* - x_i^*) + z_j^*(x_i^* - y_i^*) + z_{n+i}^*(x_i^* - y_i^*) + z_{n+j}^*(y_i^* - x_i^*)\right)$$
$$\leq 4\epsilon|\mathbf{A}_{\min}| + \frac{|\mathbf{A}_{\min}|}{\epsilon}\left(z_i^* \cdot \left(-\frac{\epsilon}{2}\right) + \epsilon^2 \cdot 2\epsilon + \epsilon^2 \cdot 2\epsilon\right) \quad\quad (8)$$
$$\leq -\left(\frac{1}{2}z_i^* - 4\epsilon^2 - 4\epsilon\right)|\mathbf{A}_{\min}|, \quad\quad (9)$$

where (8) uses the following:

- $y_i^* - x_i^* \leq -\epsilon/2$;
- $x_i^* - y_i^* \leq 2\epsilon$ (Lemma 3.2);
- $z_{n+i}^* \leq \epsilon^2$ and $z_j^* \leq \epsilon^2$; and
- $\langle \boldsymbol{x}' - \boldsymbol{x}^*, \mathbf{A}\boldsymbol{y}^* \rangle \leq \|\boldsymbol{x}' - \boldsymbol{x}^*\|_1 \|\mathbf{A}\boldsymbol{y}^*\|_\infty \leq 2|x_i^* - y_i^*||\mathbf{A}_{\min}| \leq 4\epsilon|\mathbf{A}_{\min}|$ (since $\mathbf{A}$ has negative entries).

Moreover, given that $(\boldsymbol{x}^*, \boldsymbol{y}^*, \boldsymbol{z}^*)$ is assumed to be an $\epsilon^2$-Nash equilibrium, we have

$$-u(\boldsymbol{x}', \boldsymbol{y}^*, \boldsymbol{z}^*) + u(\boldsymbol{x}^*, \boldsymbol{y}^*, \boldsymbol{z}^*) \leq \epsilon^2. \quad\quad (10)$$

(The utility of Player $\boldsymbol{x}$ is given by $-u$.) Combining (10) and (9),

$$\left(\frac{1}{2}z_i^* - 4\epsilon^2 - 4\epsilon\right)|\mathbf{A}_{\min}| \leq \epsilon^2 \quad\quad (11)$$

$$\Rightarrow \quad z_i^* \leq 8\epsilon + 10\epsilon^2 \leq 9\epsilon \text{ for } \epsilon \leq \frac{1}{10}. \quad\quad (12)$$

In summary, when $x_i^* - y_i^* \geq 0$, we have shown that $z_i^* \leq 9\epsilon$ and $z_{n+i}^* \leq \epsilon^2$. The case where $y_i^* - x_i^* \geq 0$ can be treated similarly. $\qquad\square$

We continue with the proof of Theorem 3.4, which combines Lemmas 3.2 and 3.3 to complete the CLS-hardness reduction of Section 3.1.

**Theorem 3.4.** *Given any $\epsilon^2$-Nash equilibrium $(\boldsymbol{x}^*, \boldsymbol{y}^*, \boldsymbol{z}^*)$ in the adversarial team game (2), with $\epsilon \leq 1/10$, $(\boldsymbol{y}^*, \boldsymbol{y}^*)$ is a symmetric $(21n+1)|\mathbf{A}_{\min}|\epsilon$-Nash equilibrium of the symmetric, two-player game $(\mathbf{A}, \mathbf{A})$ (that is, $\mathbf{A} = \mathbf{A}^\top$).*

*Proof.* Since $(\boldsymbol{x}^*, \boldsymbol{y}^*, \boldsymbol{z}^*)$ is an $\epsilon^2$-Nash equilibrium, we have that for any for any deviation $\boldsymbol{y}' \in \Delta^n$ of Player $\boldsymbol{y}$,

$$\langle \boldsymbol{x}^*, \mathbf{A}\boldsymbol{y}^* \rangle \leq \langle \boldsymbol{x}^*, \mathbf{A}\boldsymbol{y}' \rangle + \frac{|\mathbf{A}_{\min}|}{\epsilon}\left(\sum_{i=1}^n z_i(x_i^* - y_i') + z_{n+i}(y_i' - x_i^*)\right) + \epsilon^2. \tag{13}$$

Moreover, by considering a deviation of Player $\boldsymbol{x}$ again to $\boldsymbol{y}'$,

$$\langle \boldsymbol{x}^*, \mathbf{A}\boldsymbol{y}^* \rangle \leq \langle \boldsymbol{y}', \mathbf{A}\boldsymbol{y}^* \rangle + \frac{|\mathbf{A}_{\min}|}{\epsilon}\left(\sum_{i=1}^n z_i(y_i' - y_i^*) + z_{n+i}(y_i^* - y_i')\right) + \epsilon^2 \tag{14}$$

Adding (13) and (14), and using the fact that $\mathbf{A}$ is a symmetric matrix,

$$2\langle \boldsymbol{x}^*, \mathbf{A}\boldsymbol{y}^* \rangle \leq \langle \boldsymbol{y}', \mathbf{A}(\boldsymbol{x}^* + \boldsymbol{y}^*) \rangle + \frac{|\mathbf{A}_{\min}|}{\epsilon}\left(\sum_{i=1}^n z_i(x_i^* - y_i^*) + z_{n+i}(y_i^* - x_i^*)\right) + 2\epsilon^2$$

$$\leq \langle \boldsymbol{y}', \mathbf{A}(2\boldsymbol{y}^*) \rangle + 2\epsilon n|\mathbf{A}_{\min}| + \frac{|\mathbf{A}_{\min}|}{\epsilon}(2n \cdot 9\epsilon \cdot 2\epsilon) + 2\epsilon^2 \tag{15}$$

$$\leq 2\langle \boldsymbol{y}', \mathbf{A}\boldsymbol{y}^* \rangle + (38n+2)|\mathbf{A}_{\min}|\epsilon, \tag{16}$$

where in (15) we use Lemmas 3.2 and 3.3. Also,

$$\langle \boldsymbol{y}^*, \mathbf{A}\boldsymbol{y}^* \rangle = \langle \boldsymbol{x}^*, \mathbf{A}\boldsymbol{y}^* \rangle + \langle \boldsymbol{y}^* - \boldsymbol{x}^*, \mathbf{A}\boldsymbol{y}^* \rangle$$
$$\leq \langle \boldsymbol{x}^*, \mathbf{A}\boldsymbol{y}^* \rangle + 2\epsilon n|\mathbf{A}_{\min}|. \tag{17}$$

Finally, combining (16) and (17), we conclude that for any $\boldsymbol{y}' \in \Delta^n$,

$$\langle \boldsymbol{y}^*, \mathbf{A}\boldsymbol{y}^* \rangle \leq \langle \boldsymbol{y}', \mathbf{A}\boldsymbol{y}^* \rangle + (21n+1)|\mathbf{A}_{\min}|\epsilon.$$

This concludes the proof. $\qquad\square$

We now restate Theorem 1.2, which establishes the main complexity implication of Theorem 3.1.

**Theorem 1.2.** *Computing an $\epsilon$-Nash equilibrium in 3-player (that is, 2 vs. 1) adversarial team games is CLS-complete.*

*Proof.* CLS-hardness follows directly from Theorem 3.4 and Theorem 3.1 (due to Ghosh and Hollender [2024]). The inclusion was shown by Anagnostides et al. [2023]. $\qquad\square$

### A.2 Irrational Nash equilibria in adversarial team games

We next describe an interesting property for adversarial team games. Namely, similar to general-sum 3-player games [Nash, 1950], there exist adversarial team games that admit a unique irrational Nash equilibrium, as stated below.

**Proposition A.2** (Berthelsen and Hansen, 2019)**.** *There exists a 3-player adversarial team game with a unique Nash equilibrium that is supported on irrationals.*

Although the paper of Berthelsen and Hansen [2019] provides such an instance, no proof is given. Here, as a complement to their work, we provide a relatively simple and general way to analyze the irrational Nash equilibrium in 3-player adversarial team games.

We consider a 3-player adversarial team game in which the utility function of the adversary $u : \{1, 2\} \times \{1, 2\} \times \{1, 2\} : (\boldsymbol{x}, \boldsymbol{y}, \boldsymbol{z}) \mapsto \mathbb{R}$ reads

| $z$ \ $(\boldsymbol{x}, \boldsymbol{y})$ | (1, 1) | (1, 2) | (2, 1) | (2, 2) |
|---|---|---|---|---|
| 1 | 1 | 3 | $\frac{99}{100}$ | $-\frac{1}{100}$ |
| 2 | $\frac{9}{10}$ | $-\frac{1}{10}$ | 1 | 3 |

The proof of this result makes use of a characterization of Nash equilibria in $2 \times 2$ two-player zero-sum games, stated below; for the proof, we refer to, for example, Sun [2022, Theorem 1.2].

**Lemma A.3.** *Let* $\mathbf{A} \in \mathbb{R}^{2 \times 2}$ *such that*

$$(\mathbf{A}_{1,1} - \mathbf{A}_{1,2})(\mathbf{A}_{2,2} - \mathbf{A}_{2,1}) > 0 \text{ and } (\mathbf{A}_{1,1} - \mathbf{A}_{2,1})(\mathbf{A}_{2,2} - \mathbf{A}_{1,2}) > 0.$$

*Then, the two-player zero-sum game* $\min_{\boldsymbol{x} \in \Delta^2} \max_{\boldsymbol{z} \in \Delta^2} \langle \boldsymbol{x}, \mathbf{A}\boldsymbol{z} \rangle$ *admits a unique (exact) Nash equilibrium with value*

$$v := \frac{\mathbf{A}_{1,1}\mathbf{A}_{2,2} - \mathbf{A}_{1,2}\mathbf{A}_{2,1}}{\mathbf{A}_{1,1} - \mathbf{A}_{1,2} - \mathbf{A}_{2,1} + \mathbf{A}_{2,2}}.$$

*Furthermore, the unique Nash equilibrium* $(\boldsymbol{x}^*, \boldsymbol{z}^*)$ *satisfies*

$$\boldsymbol{x}^* = \left( \frac{\mathbf{A}_{2,2} - \mathbf{A}_{2,1}}{\mathbf{A}_{1,1} - \mathbf{A}_{1,2} - \mathbf{A}_{2,1} + \mathbf{A}_{2,2}}, \frac{\mathbf{A}_{1,1} - \mathbf{A}_{1,2}}{\mathbf{A}_{1,1} - \mathbf{A}_{1,2} - \mathbf{A}_{2,1} + \mathbf{A}_{2,2}} \right)$$

$$\boldsymbol{z}^* = \left( \frac{\mathbf{A}_{2,2} - \mathbf{A}_{1,2}}{\mathbf{A}_{1,1} - \mathbf{A}_{1,2} - \mathbf{A}_{2,1} + \mathbf{A}_{2,2}}, \frac{\mathbf{A}_{1,1} - \mathbf{A}_{2,1}}{\mathbf{A}_{1,1} - \mathbf{A}_{1,2} - \mathbf{A}_{2,1} + \mathbf{A}_{2,2}} \right).$$

*Proof of Proposition A.2.* By construction of the adversarial team game, the mixed extension of the utility can be expressed as

$$x_1 z_1 (y_1 + 3y_2) + x_1 z_2 \left( \frac{9}{10} y_1 - \frac{1}{10} y_2 \right) + x_2 z_1 \left( \frac{99}{100} y_1 - \frac{1}{100} y_2 \right) + x_2 z_2 (3y_2 + y_1).$$

Suppose that we fix $\boldsymbol{y} \in \Delta^2$. Then, Players $\boldsymbol{x}$ and $\boldsymbol{y}$ are engaged in a (two-player) zero-sum game with payoff matrix

$$\mathbf{A}(\boldsymbol{y}) := \begin{bmatrix} 1 + 2y_2 & \frac{9}{10} - y_2 \\ \frac{99}{100} - y_2 & 1 + 2y_2 \end{bmatrix}. \tag{18}$$

We now invoke Lemma A.3. Indeed, we have

$$(\mathbf{A}(\boldsymbol{y})_{1,1} - \mathbf{A}(\boldsymbol{y})_{1,2})(\mathbf{A}(\boldsymbol{y})_{2,2} - \mathbf{A}(\boldsymbol{y})_{2,1}) = \left( 3y_2 + \frac{1}{10} \right) \left( 3y_2 + \frac{1}{100} \right) > 0 \tag{19}$$

and

$$(\mathbf{A}(\boldsymbol{y})_{1,1} - \mathbf{A}(\boldsymbol{y})_{2,1})(\mathbf{A}(\boldsymbol{y})_{2,2} - \mathbf{A}(\boldsymbol{y})_{1,2}) = \left( 3y_2 + \frac{1}{100} \right) \left( 3y_2 + \frac{1}{10} \right) > 0; \tag{20}$$

that is, the precondition of Lemma A.3 is satisfied, and so the value of (18) reads

$$v(\boldsymbol{y}) = \min_{\boldsymbol{x} \in \Delta^2} \max_{\boldsymbol{z} \in \Delta^2} \langle \boldsymbol{x}, \mathbf{A}(\boldsymbol{y})\boldsymbol{z} \rangle = \frac{109 + 5890y_2 + 3000y_2^2}{110 + 6000y_2}. \tag{21}$$

It is easy to verify that $v$ is a strictly convex function in $[0, 1]$, and admits a unique minimum corresponding to $\boldsymbol{y}^* = \left( \frac{611 - 9\sqrt{3}}{600}, \frac{9\sqrt{3} - 11}{600} \right)$, which is irrational. Now, suppose that $(\boldsymbol{x}^*, \boldsymbol{y}^*, \boldsymbol{z}^*)$ is a Nash equilibrium of the adversarial team game. We will first argue that $(\boldsymbol{x}^*, \boldsymbol{z}^*)$ is the unique Nash equilibrium of $\mathbf{A}(\boldsymbol{y}^*)$. Indeed, suppose that there exists $\boldsymbol{x}' \in \Delta^2$ such that $\langle \boldsymbol{x}', \mathbf{A}(\boldsymbol{y}^*)\boldsymbol{z}^* \rangle < \langle \boldsymbol{x}^*, \mathbf{A}(\boldsymbol{y}^*)\boldsymbol{z}^* \rangle$, or equivalently, $u(\boldsymbol{x}', \boldsymbol{y}^*, \boldsymbol{z}^*) < u(\boldsymbol{x}^*, \boldsymbol{y}^*, \boldsymbol{z}^*)$; this is a contradiction since $(\boldsymbol{x}^*, \boldsymbol{y}^*, \boldsymbol{z}^*)$ is assumed to be a Nash equilibrium. Similar reasoning applies with respect to Player $\boldsymbol{z}$. Thus, $(\boldsymbol{x}^*, \boldsymbol{z}^*)$ is a Nash equilibrium of $\mathbf{A}(\boldsymbol{y}^*)$, and thereby uniquely determined by $\boldsymbol{y}^*$—by Lemma A.3 coupled with (19) and (20). Furthermore, given the value of $\boldsymbol{y}^*$, we get that $\boldsymbol{x}^* = \left( \frac{3 - \sqrt{3}}{6}, \frac{3 + \sqrt{3}}{6} \right)$ and $\boldsymbol{z}^* = \left( \frac{3 + \sqrt{3}}{6}, \frac{3 - \sqrt{3}}{6} \right)$. Now, consider the utility of Player $\boldsymbol{y}$

when playing the first action $a_1$ or the second action $a_2$; plugging in the value of $\boldsymbol{x}^*$ and $\boldsymbol{z}^*$, we have $u(\boldsymbol{x}^*, \boldsymbol{e}_1, \boldsymbol{z}^*) = \frac{578+9\sqrt{3}}{600}$ and $u(\boldsymbol{x}^*, \boldsymbol{e}_2, \boldsymbol{z}^*) = \frac{578+9\sqrt{3}}{600}$. Since $u(\boldsymbol{x}^*, \boldsymbol{e}_1, \boldsymbol{z}^*) = u(\boldsymbol{x}^*, \boldsymbol{e}_2, \boldsymbol{z}^*)$, $(\boldsymbol{x}^*, \boldsymbol{y}^*, \boldsymbol{z}^*)$ is a Nash equilibrium.

Moreover, suppose there exists another Nash equilibrium $(\boldsymbol{x}', \boldsymbol{y}', \boldsymbol{z}')$ that is different from $(\boldsymbol{x}^*, \boldsymbol{y}^*, \boldsymbol{z}^*)$. As shown above, $(\boldsymbol{x}', \boldsymbol{z}')$ is the unique equilibrium of the zero-sum game induced by $\boldsymbol{y}'$. Thus, if we have two different Nash equilibria, it implies that $\boldsymbol{y}' \neq \boldsymbol{y}^*$. We consider the following three cases:

- First, let $\boldsymbol{y}'$ be a (fully) mixed strategy. Since $\boldsymbol{x}'$ and $\boldsymbol{z}'$ forms the unique NE in of $\mathbf{A}(\boldsymbol{y}')$, we have

$$\boldsymbol{x}' = \left( \frac{1 + 300y_2'}{11 + 600y_2'}, \frac{10 + 300y_2'}{11 + 600y_2'} \right),$$

$$\boldsymbol{z}' = \left( \frac{10 + 300y_2'}{11 + 600y_2'}, \frac{1 + 300y_2'}{11 + 600y_2'} \right).$$

  Further, for Player $\boldsymbol{y}$,

$$u(\boldsymbol{x}', \boldsymbol{e}_1, \boldsymbol{z}') = \frac{1199 + 130800y_2 + 3501000y_2^2}{10(11 + 600y_2)^2},$$

$$u(\boldsymbol{x}', \boldsymbol{e}_2, \boldsymbol{z}') = \frac{589 + 196800y_2 + 5301000y_2^2}{10(11 + 600y_2)^2}.$$

  Since $\boldsymbol{y}'$ is a mixed strategy, we have $u(\boldsymbol{x}', \boldsymbol{e}_1, \boldsymbol{z}') = u(\boldsymbol{x}', \boldsymbol{e}_2, \boldsymbol{z}')$; solving the equality we get $\boldsymbol{y}' = \left( \frac{611-9\sqrt{3}}{600}, \frac{9\sqrt{3}-11}{600} \right)$, which contradicts the assumption that $\boldsymbol{y}' \neq \boldsymbol{y}^*$.

- If $\boldsymbol{y}' = (1, 0)$, we have $u(\boldsymbol{x}', \boldsymbol{e}_1, \boldsymbol{z}') = \frac{1199}{1210}$ and $u(\boldsymbol{x}', \boldsymbol{e}_2, \boldsymbol{z}') = \frac{589}{1210}$. Thus, it follows that by unilaterally deviating to play $(0, 1)$, Player $\boldsymbol{y}$ can decrease the utility of the adversary, contradicting the fact that $(\boldsymbol{x}', \boldsymbol{y}', \boldsymbol{z}')$ is a Nash equilibrium.

- Finally, suppose that $\boldsymbol{y}' = (0, 1)$. Similarly to the second case, we get $u(\boldsymbol{x}', \boldsymbol{e}_1, \boldsymbol{z}') < u(\boldsymbol{x}', \boldsymbol{e}_2, \boldsymbol{z}')$, which is a contradiction.

Thus, we conclude that $(\boldsymbol{x}^*, \boldsymbol{y}^*, \boldsymbol{z}^*)$ is the unique Nash equilibrium of the 3-player adversarial team game defined above, completing the proof. $\square$

A natural question arising from Proposition A.2 concerns the complexity of determining whether an adversarial team game admits a unique Nash equilibrium. Theorem 3.5—that was presented earlier in the main body—establishes NP-hardness for a version of that problem that accounts for approximate Nash equilibria.

### A.3 Proofs from Section 4.1

We continue with the proofs from Section 4.1. We first apply Brouwer's fixed point theorem to show that symmetric min-max optimization problems always admit a symmetric equilibrium.

**Lemma A.4.** *Let $\mathcal{X}$ be a convex and compact set. Then, any $L$-smooth, antisymmetric function (Definition 1.4) $f : \mathcal{X} \times \mathcal{X} \to \mathbb{R}$ admits a symmetric first-order Nash equilibrium $(\boldsymbol{x}^*, \boldsymbol{x}^*)$.*

*Proof.* We define the function $M : \mathcal{X} \to \mathcal{X}$ to be

$$M(\boldsymbol{x}') := \prod_{\mathcal{X}} \left[ \boldsymbol{x}' - \nabla_{\boldsymbol{x}} f(\boldsymbol{x}, \boldsymbol{y}) \Big|_{(\boldsymbol{x}, \boldsymbol{y}) = (\boldsymbol{x}', \boldsymbol{x}')} \right]. \tag{22}$$

Given that $f$ is $L$-smooth, we conclude that $M(\boldsymbol{x}')$ is $(L+1)$-Lipschitz, hence continuous. Therefore, from Brouwer's fixed point theorem, there exists an $\boldsymbol{x}^*$ so that $M(\boldsymbol{x}^*) = \boldsymbol{x}^*$. Moreover, the symmetry

of $f$ implies that $\nabla_{\boldsymbol{y}} f(\boldsymbol{x}, \boldsymbol{y})\big|_{(\boldsymbol{x},\boldsymbol{y})=(\boldsymbol{w},\boldsymbol{w})} = -\nabla_{\boldsymbol{x}} f(\boldsymbol{x}, \boldsymbol{y})\big|_{(\boldsymbol{x},\boldsymbol{y})=(\boldsymbol{w},\boldsymbol{w})}$ for all $\boldsymbol{w} \in \mathcal{X}$; as a result,

$$
\begin{aligned}
\boldsymbol{x}^* &= \textstyle\prod_{\mathcal{X}} \left[\boldsymbol{x}^* - \nabla_{\boldsymbol{x}} f(\boldsymbol{x}, \boldsymbol{y})\big|_{(\boldsymbol{x},\boldsymbol{y})=(\boldsymbol{x}^*,\boldsymbol{x}^*)}\right] \\
&= \textstyle\prod_{\mathcal{X}} \left[\boldsymbol{x}^* + \nabla_{\boldsymbol{y}} f(\boldsymbol{x}, \boldsymbol{y})\big|_{(\boldsymbol{x},\boldsymbol{y})=(\boldsymbol{x}^*,\boldsymbol{x}^*)}\right].
\end{aligned}
$$

Therefore, $(\boldsymbol{x}^*, \boldsymbol{x}^*)$ is a first-order Nash equilibrium of the symmetric min-max problem with function $f$. □

We now present the proof of Lemma 4.1

*Proof of Lemma 4.1.* We first define the function (as in Lemma A.4) $M : \mathcal{X} \to \mathcal{X}$ as

$$
M(\boldsymbol{x}') \coloneqq \prod_{\mathcal{X}} \left[\boldsymbol{x}' - \nabla_{\boldsymbol{x}} f(\boldsymbol{x}, \boldsymbol{y})\big|_{(\boldsymbol{x},\boldsymbol{y})=(\boldsymbol{x}',\boldsymbol{x}')}\right],
$$

where we recall that $\Pi$ is the projection operator on $\mathcal{X}$. Assuming that the input function $f$ is $L$-smooth, it follows that $M(\boldsymbol{x}')$ is $(L+1)$-Lipschitz. Furthermore, projecting on the polytope $\mathcal{X}$ takes polynomial time, and so $M$ is polynomial-time computable. As a result, we can use Etessami and Yannakakis [2010, Proposition 2, part 2] (see also Fearnley et al. [2023, Proposition D.1]), where it was shown that finding an $\epsilon$-approximate fixed point of a Brouwer function that is efficiently computable and continuous, when the domain is a bounded polytope, lies in PPAD. □

We proceed with the proof of Theorem 4.2

*Proof of Theorem 4.2.* We $P$-time reduce the problem of finding approximate symmetric NE in two-player symmetric games to SYMGDAFIXEDPOINT. Given any two-player symmetric game with payoff matrices $(\mathbf{R}, \mathbf{R}^\top)$ of size $n \times n$, we set

$$
\mathbf{A} \coloneqq \frac{1}{2}\left(\mathbf{R} + \mathbf{R}^\top\right) \text{ (symmetric matrix) and } \mathbf{C} \coloneqq \frac{1}{2}\left(\mathbf{R} - \mathbf{R}^\top\right) \text{ (skew-symmetric matrix).} \quad (23)
$$

We define the *quadratic*, antisymmetric function

$$
f(\boldsymbol{x}, \boldsymbol{y}) \coloneqq \frac{1}{2}\langle \boldsymbol{y}, \mathbf{A}\boldsymbol{y} \rangle - \frac{1}{2}\langle \boldsymbol{x}, \mathbf{A}\boldsymbol{x} \rangle + \langle \boldsymbol{y}, \mathbf{C}\boldsymbol{x} \rangle \quad (24)
$$

with domain $\Delta^n \times \Delta^n$. Indeed, to see that $f$ is antisymmetric, one can observe that

$$
f(\boldsymbol{y}, \boldsymbol{x}) = \frac{1}{2}\langle \boldsymbol{x}, \mathbf{A}\boldsymbol{x} \rangle - \frac{1}{2}\langle \boldsymbol{y}, \mathbf{A}\boldsymbol{y} \rangle + \langle \boldsymbol{x}, \mathbf{C}\boldsymbol{y} \rangle = \frac{1}{2}\langle \boldsymbol{x}, \mathbf{A}\boldsymbol{x} \rangle - \frac{1}{2}\langle \boldsymbol{y}, \mathbf{A}\boldsymbol{y} \rangle - \langle \boldsymbol{y}, \mathbf{C}^\top \boldsymbol{x} \rangle = -f(\boldsymbol{x}, \boldsymbol{y}).
$$

Assuming that all entries of $\mathbf{R}$ lie in $[-1, 1]$, it follows that the singular values of $\mathbf{A}$ and $\mathbf{C}$ are bounded by $n$. As a result $f$ and $\nabla_{\boldsymbol{x}} f = -\mathbf{A}\boldsymbol{x} - \mathbf{C}\boldsymbol{y}, \nabla_{\boldsymbol{y}} f = \mathbf{A}\boldsymbol{y} + \mathbf{C}\boldsymbol{x}$ are polynomial time computable and continuous, and $\nabla_{\boldsymbol{x}} f, \nabla_{\boldsymbol{y}} f$ are $L$-Lipschitz for $L \leq 2n$, thus $f$ is $4n$-smooth.

We assume $\boldsymbol{x}$ is the minimizer and $\boldsymbol{y}$ is the maximizer, and let $(\boldsymbol{x}^*, \boldsymbol{x}^*)$ be an $\epsilon$-approximate fixed point of GDA. We shall show that $(\boldsymbol{x}^*, \boldsymbol{x}^*)$ is an $4n\epsilon$-approximate NE of the symmetric two-player game $(\mathbf{R}, \mathbf{R}^\top)$. Since $(\boldsymbol{x}^*, \boldsymbol{x}^*)$ is an $\epsilon$-approximate fixed point of GDA, we can use Lemma A.5 and obtain the following variational inequalities:

$$
\max_{\boldsymbol{x}^*+\boldsymbol{\delta}\in\Delta^n, \|\boldsymbol{\delta}\|_2 \leq 1} \boldsymbol{\delta}^\top (\mathbf{A}\boldsymbol{x}^* + \mathbf{C}\boldsymbol{x}^*) \leq \epsilon(2n+1),
$$

implying that (since the diameter of $\Delta^n$ is $\sqrt{2}$ in $\ell_2$)

$$
\langle \boldsymbol{x} - \boldsymbol{x}^*, (\mathbf{A}+\mathbf{C})\boldsymbol{x}^* \rangle \leq \sqrt{2}\epsilon(2n+1) \text{ for any } \boldsymbol{x} \in \Delta^n. \quad \text{(VI for NE)}
$$

Now, we observe that (VI for NE) implies that $(\boldsymbol{x}^*, \boldsymbol{x}^*)$ is a $\sqrt{2}\epsilon(2n+1)$-approximate symmetric NE in the two-player symmetric game with payoff matrices $(\mathbf{A}+\mathbf{C}, \mathbf{A}-\mathbf{C})$ (recall Definition (2.1)). Since $\sqrt{2}\epsilon(2n+1) \leq 4n\epsilon$ for $n \geq 2$, our claim follows.

By Theorem 2.2 and Lemma 4.1, we conclude that SYMGDAFIXEDPOINT is PPAD-complete, even for quadratic functions that are $O(n)$-smooth, $O(n)$-Lipschitz and $\epsilon \leq 1/n^{1+c}$, for any $c > 0$. □

We next state a standard lemma that connects first-order optimality with the fixed-point gap of gradient ascent.

**Lemma A.5** ([Ghadimi and Lan, 2016], Lemma 3 for $c = 1$). *Let $f(\boldsymbol{x})$ be an $L$-smooth function in $\boldsymbol{x} \in \Delta^n$. Define the gradient mapping*

$$G(\boldsymbol{x}) := \prod_{\Delta^n} \{\boldsymbol{x} + \nabla_{\boldsymbol{x}} f(\boldsymbol{x})\} - \boldsymbol{x}.$$

*If $\|G(\boldsymbol{x}^*)\|_2 \leq \epsilon$, that is, $\boldsymbol{x}^*$ is an $\epsilon$-approximate fixed point of gradient ascent with stepsize equal to one, then*

$$\max_{\boldsymbol{x}^* + \boldsymbol{\delta} \in \Delta^n, \|\boldsymbol{\delta}\|_2 \leq 1} \boldsymbol{\delta}^\top \nabla_{\boldsymbol{x}} f(\boldsymbol{x}^*) \leq \epsilon(L + 1).$$

Similarly, the next lemma makes such a connection for min-max optimization problems with coupled constraints; it is mostly extracted from Daskalakis et al. [2021, Section B.2].

**Lemma A.6.** *Let $f(\boldsymbol{x}, \boldsymbol{y})$ be a $G$-Lipschitz, $L$-smooth function defined in some polytope domain $\mathcal{D} \subseteq \Delta^n \times \Delta^n$ of diameter $D$. Define the mapping*

$$G(\boldsymbol{x}, \boldsymbol{y}) := \prod_{\mathcal{D}} \{\boldsymbol{x} - \nabla_{\boldsymbol{x}} f(\boldsymbol{x}, \boldsymbol{y}), \boldsymbol{y} + \nabla_{\boldsymbol{y}} f(\boldsymbol{x}, \boldsymbol{y})\} - (\boldsymbol{x}, \boldsymbol{y}).$$

*If $\|G(\boldsymbol{x}^*, \boldsymbol{y}^*)\|_2 \leq \epsilon$, that is, $(\boldsymbol{x}^*, \boldsymbol{y}^*)$ is an $\epsilon$-approximate fixed point of (the safe version) of GDA with stepsize equal to one, then*

$$\langle \boldsymbol{x} - \boldsymbol{x}^*, \nabla_{\boldsymbol{x}} f(\boldsymbol{x}^*, \boldsymbol{y}^*) \rangle \geq -\sqrt{\epsilon} K \text{ for } \boldsymbol{x} \in \mathcal{D}(\boldsymbol{y}^*) \text{ and } \langle \boldsymbol{y} - \boldsymbol{y}^*, \nabla_{\boldsymbol{y}} f(\boldsymbol{x}^*, \boldsymbol{y}^*) \rangle \leq \sqrt{\epsilon} K \text{ for } \boldsymbol{y} \in \mathcal{D}(\boldsymbol{x}^*),$$

*where $\mathcal{D}(\boldsymbol{x}^*) = \{\boldsymbol{y} : (\boldsymbol{x}^*, \boldsymbol{y}) \in \mathcal{D}\}$, $\mathcal{D}(\boldsymbol{y}^*) = \{\boldsymbol{x} : (\boldsymbol{x}, \boldsymbol{y}^*) \in \mathcal{D}\}$ and $K = (L + 1)\sqrt{(G + 4\sqrt{2})}$.*

*Proof.* Let $(\boldsymbol{x}_\Delta, \boldsymbol{y}_\Delta) = (\boldsymbol{x}^* - \nabla_{\boldsymbol{x}} f(\boldsymbol{x}^*, \boldsymbol{y}^*), \boldsymbol{y}^* + \nabla_{\boldsymbol{y}} f(\boldsymbol{x}^*, \boldsymbol{y}^*))$. In Daskalakis et al. [2021, Claim B.2], it was shown that for all $(\boldsymbol{x}, \boldsymbol{y}) \in \mathcal{D}$, we have

$$\langle (\boldsymbol{x}_\Delta, \boldsymbol{y}_\Delta) - (\boldsymbol{x}^*, \boldsymbol{y}^*), (\boldsymbol{x}, \boldsymbol{y}) - (\boldsymbol{x}^*, \boldsymbol{y}^*) \rangle \leq (G + 2D)\epsilon.$$

Using the above inequality, it was concluded that $(\boldsymbol{x}^*, \boldsymbol{y}^*)$ is an approximate fixed point of the "unsafe" version of GDA; specifically,

$$\left\| \boldsymbol{x}^* - \prod_{\mathcal{D}(\boldsymbol{y}^*)} \{\boldsymbol{x}^* - \nabla_{\boldsymbol{x}} f(\boldsymbol{x}^*, \boldsymbol{y}^*)\} \right\| \leq \sqrt{(G + 2D)\epsilon}$$

and

$$\left\| \boldsymbol{y}^* - \prod_{\mathcal{D}(\boldsymbol{x}^*)} \{\boldsymbol{y}^* + \nabla_{\boldsymbol{y}} f(\boldsymbol{x}^*, \boldsymbol{y}^*)\} \right\| \leq \sqrt{(G + 2D)\epsilon}.$$

We now use Lemma A.5 for both inequalities above, together the fact that $D = 2\sqrt{2}$, to conclude that

$$\langle \boldsymbol{x} - \boldsymbol{x}^*, \nabla_{\boldsymbol{x}} f(\boldsymbol{x}^*, \boldsymbol{y}^*) \rangle \geq -\sqrt{(G + 4\sqrt{2})\epsilon}(L + 1) \text{ for } \boldsymbol{x} \in \mathcal{D}(\boldsymbol{y}^*),$$

and

$$\langle \boldsymbol{y} - \boldsymbol{y}^*, \nabla_{\boldsymbol{y}} f(\boldsymbol{x}^*, \boldsymbol{y}^*) \rangle \leq \sqrt{(G + 4\sqrt{2})\epsilon}(L + 1) \text{ for } \boldsymbol{y} \in \mathcal{D}(\boldsymbol{x}^*).$$

$\square$

We proceed to establish Theorem 4.4.

*Proof of Theorem 4.4.* The proof follows similar steps with Theorem 4.2, namely, we $P$-time reduce the problem of finding approximate symmetric NE in two-player symmetric games to GDAFIXEDPOINT with coupled domains. Given a two-player symmetric game with payoff matrices $(\mathbf{R}, \mathbf{R}^\top)$ of size $n \times n$, we set $\mathbf{A} := \frac{1}{2}(\mathbf{R} + \mathbf{R}^\top)$, $\mathbf{C} := \frac{1}{2}(\mathbf{R} - \mathbf{R}^\top)$ and define again the quadratic, antisymmetric function

$$f(\boldsymbol{x}, \boldsymbol{y}) := \frac{1}{2}\langle \boldsymbol{y}, \mathbf{A}\boldsymbol{y} \rangle - \frac{1}{2}\langle \boldsymbol{x}, \mathbf{A}\boldsymbol{x} \rangle + \langle \boldsymbol{y}, \mathbf{C}\boldsymbol{x} \rangle.$$

Moreover, given a parameter $\delta > 0$ (to be specified shortly), we define the joint domain of $f$ to be

$$\mathcal{D} := \{(\boldsymbol{x}, \boldsymbol{y}) \in \Delta^n \times \Delta^n : -\delta \leq x_i - y_i \leq \delta \text{ for all } i \in [n]\}. \qquad \text{(joint Domain)}$$

Let $(\boldsymbol{x}^*, \boldsymbol{y}^*)$ be an $\epsilon$-approximate fixed point of GDA. We will show that $\left(\frac{\boldsymbol{x}^*+\boldsymbol{y}^*}{2}, \frac{\boldsymbol{x}^*+\boldsymbol{y}^*}{2}\right)$ is an $O(\epsilon^{1/4})$-approximate (symmetric) NE of the game $(\mathbf{R}, \mathbf{R}^\top)$ for an appropriate choice of $\delta$.

We set $\mathcal{D}(\boldsymbol{x}^*) = \{\boldsymbol{y} : (\boldsymbol{x}^*, \boldsymbol{y}) \in \mathcal{D}\}$ and $\mathcal{D}(\boldsymbol{y}^*) = \{\boldsymbol{x} : (\boldsymbol{x}, \boldsymbol{y}^*) \in \mathcal{D}\}$. In words, $\mathcal{D}(\boldsymbol{x}^*)$ and $\mathcal{D}(\boldsymbol{y}^*)$ capture the allowed deviations for $\boldsymbol{y}$ and $\boldsymbol{x}$ respectively. It also holds that $f$ is $G$-Lipschitz continuous with $G = 4n$ and also $4n$-smooth (using the same reasoning as in Theorem 4.2).

Since $(\boldsymbol{x}^*, \boldsymbol{y}^*)$ is an $\epsilon$-approximate fixed point of GDA, using Lemma A.6, the following VIs must hold for some positive constant $K < 10$ and $n$ sufficiently large:

$$\begin{aligned} \langle \boldsymbol{x} - \boldsymbol{x}^*, -\mathbf{A}\boldsymbol{x}^* + \mathbf{C}^\top \boldsymbol{y}^* \rangle &\geq -Kn^{3/2}\sqrt{\epsilon} \text{ for any } \boldsymbol{x} \in \mathcal{D}(\boldsymbol{y}^*) \text{ and} \\ \langle \boldsymbol{y} - \boldsymbol{y}^*, \mathbf{A}\boldsymbol{y}^* + \mathbf{C}\boldsymbol{x}^* \rangle &\leq Kn^{3/2}\sqrt{\epsilon} \text{ for any } \boldsymbol{y} \in \mathcal{D}(\boldsymbol{x}^*). \end{aligned} \qquad (25)$$

Let $\overline{\mathcal{D}} = \left\{\boldsymbol{z} \in \Delta^n : \left\|\boldsymbol{z} - \frac{\boldsymbol{x}^*+\boldsymbol{y}^*}{2}\right\|_\infty \leq \frac{\delta}{2}\right\}$. By triangle inequality, it follows that $\overline{\mathcal{D}} \subseteq \mathcal{D}(\boldsymbol{y}^*) \cap \mathcal{D}(\boldsymbol{x}^*)$. We express the VIs of (25) using a single variable $\boldsymbol{z}$ and common deviation domain:

$$\langle \boldsymbol{z} - \boldsymbol{x}^*, -\mathbf{A}\boldsymbol{x}^* + \mathbf{C}^\top \boldsymbol{y}^* \rangle \geq -Kn^{3/2}\sqrt{\epsilon} \text{ and } \langle \boldsymbol{z} - \boldsymbol{y}^*, \mathbf{A}\boldsymbol{y}^* + \mathbf{C}\boldsymbol{x}^* \rangle \leq Kn^{3/2}\sqrt{\epsilon} \text{ for any } \boldsymbol{z} \in \overline{\mathcal{D}}.$$

Multiplying the first inequality by $-1/2$ and the second with $1/2$ and adding them up gives

$$\left\langle \boldsymbol{z} - \frac{\boldsymbol{x}^* + \boldsymbol{y}^*}{2}, (\mathbf{A} + \mathbf{C})\frac{\boldsymbol{x}^* + \boldsymbol{y}^*}{2} \right\rangle \leq \frac{1}{4}\langle \boldsymbol{x}^* - \boldsymbol{y}^*, \mathbf{A}(\boldsymbol{x}^* - \boldsymbol{y}^*)\rangle + Kn^{3/2}\sqrt{\epsilon}. \qquad (26)$$

Since $\boldsymbol{x}^*, \boldsymbol{y}^* \in \mathcal{D}$, it follows that $\langle \boldsymbol{x}^* - \boldsymbol{y}^*, \mathbf{A}(\boldsymbol{x}^* - \boldsymbol{y}^*)\rangle \leq n\|\boldsymbol{x}^* - \boldsymbol{y}^*\|_2^2 \leq n^2\delta^2$. Combining this fact with (26), we conclude that

$$\left\langle \boldsymbol{z} - \frac{\boldsymbol{x}^* + \boldsymbol{y}^*}{2}, (\mathbf{A} + \mathbf{C})\frac{\boldsymbol{x}^* + \boldsymbol{y}^*}{2} \right\rangle \leq n^2\delta^2 + Kn^{3/2}\sqrt{\epsilon} \text{ for any } \boldsymbol{z} \in \overline{\mathcal{D}}. \qquad \text{(VImedian)}$$

(VImedian) shows that by deviating from $\left(\frac{\boldsymbol{x}^*+\boldsymbol{y}^*}{2}, \frac{\boldsymbol{x}^*+\boldsymbol{y}^*}{2}\right)$ to some $\boldsymbol{z}$ in $\overline{\mathcal{D}}$, the payoff cannot increase by more than $\left(n^2\delta^2 + Kn^{3/2}\sqrt{\epsilon}\right)$ in the two-player symmetric game with matrices $(\mathbf{R}, \mathbf{R}^\top)$.

We consider any pure strategy $\boldsymbol{e}_j$ for $j \in [n]$. If $\left\|\boldsymbol{e}_j - \frac{\boldsymbol{x}^*+\boldsymbol{y}^*}{2}\right\|_\infty \leq \frac{\delta}{2}$ then $\boldsymbol{e}_j \in \overline{\mathcal{D}}$ and it is captured by (VImedian). Suppose that $\left\|\boldsymbol{e}_j - \frac{\boldsymbol{x}^*+\boldsymbol{y}^*}{2}\right\|_\infty > \frac{\delta}{2}$ and consider the point $\boldsymbol{z}' \in \overline{\mathcal{D}}$ on the line segment between $\boldsymbol{e}_j$ and $\frac{\boldsymbol{x}^*+\boldsymbol{y}^*}{2}$ that intersects the boundary of $\overline{\mathcal{D}}$. It holds that $\boldsymbol{e}_j - \frac{\boldsymbol{x}^*+\boldsymbol{y}^*}{2} = c\left(\boldsymbol{z}' - \frac{\boldsymbol{x}^*+\boldsymbol{y}^*}{2}\right)$ for some positive $c \leq \frac{2}{\delta}$ (it cannot be larger because otherwise the infinity norm of the difference between $\boldsymbol{e}_j$ and $\frac{\boldsymbol{x}^*+\boldsymbol{y}^*}{2}$ would exceed one, which is impossible as they both belong to $\Delta^n$). Therefore,

$$\left\langle \boldsymbol{e}_j - \frac{\boldsymbol{x}^* + \boldsymbol{y}^*}{2}, (\mathbf{A} + \mathbf{C})\frac{\boldsymbol{x}^* + \boldsymbol{y}^*}{2} \right\rangle \leq 2n^2\delta + \frac{2Kn^{3/2}\sqrt{\epsilon}}{\delta} \text{ for any pure strategy } j. \qquad (27)$$

From (27), we conclude that $\frac{\boldsymbol{x}^*+\boldsymbol{y}^*}{2}$ is $\left(2n^2\delta + \frac{2Kn^{3/2}\sqrt{\epsilon}}{\delta}\right)$-approximate NE of the symmetric two-player game $(\mathbf{R}, \mathbf{R}^\top)$. We choose $\delta = \epsilon^{1/4}n^{-1/4}$ and we get that $\frac{\boldsymbol{x}^*+\boldsymbol{y}^*}{2}$ is an $O(n^{7/4}\epsilon^{1/4})$-approximate NE for $(\mathbf{R}, \mathbf{R}^\top)$, and thus the hardness result holds for $\epsilon$ of order $O\left(\frac{1}{n^{7+c}}\right)$ for any constant $c > 0$. We note that if instead of an $\epsilon$-approximate fixed point of GDA, we were given an $\epsilon$-approximate first-order NE, then the hardness result would hold for any $\epsilon$ of order $\frac{1}{n^c}$ with $c > 0$. $\qquad \square$

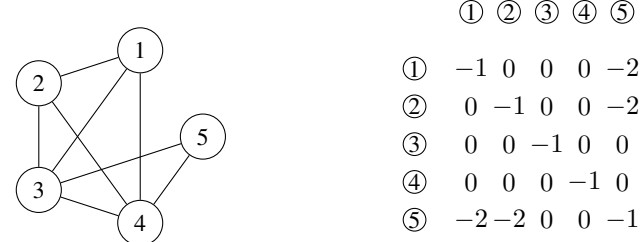

Figure 1: An example of matrix $\mathbf{A} = \mathbf{A}(G)$ (right) for graph $G$ (left).

## A.4 Proofs from Section 4.2

The main result we want to show is restated below.

**Theorem 1.6.** *For a symmetric min-max optimization problem, constants $c_1, c_2 > 0$, and $\epsilon = n^{-c_1}$, it is* NP-*hard to distinguish between the following two cases under the promise that one of them holds:*

- *any $\epsilon$-first-order Nash equilibrium $(\boldsymbol{x}^*, \boldsymbol{y}^*)$ satisfies $\|\boldsymbol{x}^* - \boldsymbol{y}^*\| \le n^{-c_2}$, and*
- *there is an $\epsilon$-first-order Nash equilibrium $(\boldsymbol{x}^*, \boldsymbol{y}^*)$ such that $\|\boldsymbol{x}^* - \boldsymbol{y}^*\| \ge \Omega(1)$.*

Our reduction builds on the hardness result of McLennan and Tourky [2010]—in turn based on earlier work by Gilboa and Zemel [1989], Conitzer and Sandholm [2008]—which we significantly refine in order to account for poly$(1/n)$-Nash equilibria. We begin by describing their basic approach. Let $G = ([n], E)$ be an $n$-node, undirected, unweighted graph, and construct

$$\mathbf{A}_{i,j} = \begin{cases} -1 & \text{if } i = j, \\ 0 & \text{if } \{i, j\} \in E, \\ -2 & \text{otherwise.} \end{cases} \tag{28}$$

(Figure 1 depicts an illustrative example.) Based on this matrix, McLennan and Tourky [2010] consider the symmetric, identical-payoff, two-player game $(\mathbf{A}, \mathbf{A})$—by construction, $\mathbf{A} = \mathbf{A}^\top$, and so this game is indeed symmetric. They were able to show the following key property.

**Lemma A.7** ([McLennan and Tourky, 2010]). *Let $C_k \subseteq [n]$ be a maximum clique of $G$ with size $k$ and $\boldsymbol{x}^* = \frac{1}{k} \sum_{i \in C_k} \boldsymbol{e}_i$. Then, $(\boldsymbol{x}^*, \boldsymbol{x}^*)$ is a Nash equilibrium of $(\mathbf{A}, \mathbf{A})$ that attains value $-\frac{1}{k}$. Furthermore, any symmetric Nash equilibrium not in the form described above has value at most $-\frac{1}{k-1}$.*

The idea now is to construct a new symmetric, identical-payoff game $(\mathbf{B}, \mathbf{B})$, for

$$\mathbf{B} := \begin{bmatrix} \mathbf{A}_{1,1} & \cdots & \mathbf{A}_{1,n} & r \\ \vdots & \ddots & \vdots & \vdots \\ \mathbf{A}_{n,1} & \cdots & \mathbf{A}_{n,n} & r \\ r & \cdots & r & V \end{bmatrix}, \tag{29}$$

where $V := -\frac{1}{k}$ and $r = \frac{1}{2}\left(-\frac{1}{k} - \frac{1}{k-1}\right) = -\frac{2k-1}{2(k-1)k}$. Coupled with Lemma A.7, this new game yields the following NP-hardness result.

**Theorem A.8** ([McLennan and Tourky, 2010]). *It is* NP-*hard to determine whether a symmetric, identical-payoff, two-player game has a unique symmetric Nash equilibrium.*

Our goal here is to prove a stronger result, Theorem 4.7, that characterizes the set of $\epsilon$-Nash equilibria even for $\epsilon = 1/n^c$; this will form the basis for our hardness result in min-max optimization and adversarial team games. To do so, we first derive some basic properties of game (29).

Game (29) always admits the trivial (symmetric) Nash equilibrium $(\boldsymbol{e}_{n+1}, \boldsymbol{e}_{n+1})$. Now, consider any symmetric Nash equilibrium $(\boldsymbol{x}^*, \boldsymbol{x}^*)$ with $x_{n+1}^* \ne 1$. If $x_{n+1}^* = 0$, it follows that $(\boldsymbol{x}^*_{[i \cdots n]}, \boldsymbol{x}^*_{[i \cdots n]})$ is a Nash equilibrium of $(\mathbf{A}, \mathbf{A})$, which in turn implies that $G$ admits a clique of size $k$; this follows from Lemma A.7, together with the fact that $-1/(k-1) < r < -1/k$.

We now analyze the case where $x_{n+1}^* \in (0, 1)$. It then follows that $(\boldsymbol{x}^*_{[1 \cdots n]}/(1-x_{n+1}^*), \boldsymbol{x}^*_{[1 \cdots n]}/(1-x_{n+1}^*))$ is a (symmetric) Nash equilibrium of $(\mathbf{A}, \mathbf{A})$. Furthermore, the utility of playing action $a_{n+1}$ is

$(1 - x^*_{n+1})r + x^*_{n+1}V > r$. By Lemma A.7, it follows that $(\boldsymbol{x}^*_{[1\cdots n]}/(1-x^*_{n+1}), \boldsymbol{x}^*_{[1\cdots n]}/(1-x^*_{n+1}))$ has a value of $V$ and $G$ admits a clique of size $k$. As a result, the utility of playing any action $a_i$, with $i \in \text{supp}(\boldsymbol{x}^*)$ and $i \neq n+1$, is $(1-x^*_{n+1})V + x^*_{n+1}r$. At the same time, the utility of playing action $a_{n+1}$ reads $(1 - x^*_{n+1})r + x^*_{n+1}V$. Equating those two quantities, it follows that $x^*_{n+1} = 1/2$.

In summary, $G$ contains a clique of size $k$ if and only if game (29) admits a unique symmetric Nash equilibrium, which implies Theorem A.8. What is more, we have shown a stronger property. Namely, any symmetric Nash equilibrium of $(\mathbf{B}, \mathbf{B})$ has to be in one of the following forms:

1. $(\boldsymbol{x}^*, \boldsymbol{x}^*)$ with $\boldsymbol{x}^* := \boldsymbol{e}_{n+1}$;
2. $(\boldsymbol{x}^*, \boldsymbol{x}^*)$ with $\boldsymbol{x}^* := \frac{1}{k}\sum_{i \in C_k} \boldsymbol{e}_i$, where $C_k \subseteq [n]$ is a clique in $G$ of size $k$;
3. $(\boldsymbol{x}^*, \boldsymbol{x}^*)$ with $\boldsymbol{x}^* := \frac{1}{2}\boldsymbol{e}_{n+1} + \frac{1}{2k}\sum_{i \in C_k} \boldsymbol{e}_i$, where $C_k \subseteq [n]$ is a clique in $G$ of size $k$.

In particular, the equilibria in Items 2 or 3—which exist iff $G$ contains a clique of size $k$—are always far from the one in Item 1. However, this characterization only applies to exact Nash equilibria. In any two-player game $\Gamma$, when $\epsilon$ is sufficiently small with $\log(1/\epsilon) \leq \text{poly}(|\Gamma|)$, Etessami and Yannakakis [2010] have shown that any $\epsilon$-Nash equilibrium is within $\ell_1$-distance $\delta$ from an exact one, and so the above characterization can be applied; unfortunately, this does not apply (for general games) in the regime we are interested, namely $\epsilon = \text{poly}(1/n)$.

We address this challenge by refining the result of McLennan and Tourky [2010]. Our main result, which forms the basis for Theorem 1.6 and Theorem 3.5, is recalled below.

**Theorem A.9.** *For symmetric, identical-interest, two-player games, constants $c_1, c_2 > 0$, and $\epsilon = n^{-c_1}$, it is* NP-*hard to distinguish between the following two cases:*

- *any two symmetric $\epsilon$-Nash equilibria have $\ell_1$-distance at most $n^{-c_2}$, and*
- *there are two symmetric $\epsilon$-Nash equilibria that have $\ell_1$-distance $\Omega(1)$.*

Our reduction proceeds similarly, but defines $\overline{\mathbf{A}}$ to be the adjacency matrix of $G$ with $\delta \in (0,1)$ in each diagonal entry. Using $\overline{\mathbf{A}}$, we show that we can refine Lemma A.7 of McLennan and Tourky [2010]. Before we state the key property we prove in Lemma A.11, we recall the following definition.

**Definition A.10** (Well-supported NE). A symmetric strategy profile $(\boldsymbol{x}, \boldsymbol{x})$ is an $\epsilon$-*well-supported* Nash equilibrium of the symmetric, identical-payoff game $(\overline{\mathbf{A}}, \overline{\mathbf{A}})$ if for all $i \in [n]$,

$$x_i > 0 \implies (\overline{\mathbf{A}}\boldsymbol{x})_i \geq \max_{j \in [n]} (\overline{\mathbf{A}}\boldsymbol{x})_j - \epsilon.$$

**Lemma A.11.** *Suppose that the maximum clique in $G$ is of size $k$. For any symmetric $\epsilon$-well-supported NE $(\hat{\boldsymbol{x}}, \hat{\boldsymbol{x}})$ of $(\overline{\mathbf{A}}, \overline{\mathbf{A}})$ not supported on a clique of size $k$, we have $u(\hat{\boldsymbol{x}}, \hat{\boldsymbol{x}}) \leq 1 - \frac{1}{k} + \frac{\delta}{k} - \frac{2\delta}{n^2 k^4} + 2\epsilon$.*

Equipped with this property, we will see shortly that a similar argument to the one described earlier concerning game (29) establishes Theorem 4.7. We proceed now with the proof of Theorem 1.6.

*Proof of Theorem 1.6.* It suffices to consider the antisymmetric function $f(\boldsymbol{x}, \boldsymbol{y}) := \boldsymbol{y}^\top \mathbf{B}\boldsymbol{y} - \boldsymbol{x}^\top \mathbf{B}\boldsymbol{x}$, where symmetric matrix $\mathbf{B}$ is defined as in (29), using our new matrix $\overline{\mathbf{A}}$ instead of $\mathbf{A}$ (see (32)). Any $\epsilon$-first-order Nash equilibrium $(\boldsymbol{x}^*, \boldsymbol{y}^*)$ of this (separable) min-max optimization problem induces, two symmetric $\epsilon$-Nash equilibria—namely, $(\boldsymbol{x}^*, \boldsymbol{x}^*)$ and $(\boldsymbol{y}^*, \boldsymbol{y}^*)$—in the symmetric, identical-interest, game $(\mathbf{B}, \mathbf{B})$. Using Theorem 4.7, the claim follows. $\qquad \square$

In what follows, our goal is to establish Theorem 4.7, which forms the basis for Theorems 1.6 and 3.5.

We consider the symmetric game $(\overline{\mathbf{A}}, \overline{\mathbf{A}}^\top)$, where $\overline{\mathbf{A}} \in \mathbb{R}^{n \times n}$ is a symmetric matrix. In particular, since $\overline{\mathbf{A}} = \overline{\mathbf{A}}^\top$, the two players in the game share the same payoff matrix. The payoff matrix $\overline{\mathbf{A}}$ is constructed based on an underlying graph $G = ([n], E)$ and a parameter $\delta \in (0, 1)$ as follows:

$$\overline{\mathbf{A}}_{i,j} = \begin{cases} \delta & \text{if } i = j, \\ 1 & \text{if } (i,j) \in E, \\ 0 & \text{otherwise.} \end{cases}$$

**Lemma A.12.** *Let $(\hat{\boldsymbol{x}}, \hat{\boldsymbol{x}})$ be an $\epsilon$-well-supported NE of the game where $\hat{\boldsymbol{x}}$ is supported on a max clique of size $k$, denoted as $C_k$, then the value $u(\hat{\boldsymbol{x}}, \hat{\boldsymbol{x}})$ is at least $1 - \frac{1}{k} + \frac{\delta}{k} - (\frac{k-\delta}{1-\delta})\epsilon$ and $\|\hat{\boldsymbol{x}} - \boldsymbol{x}^*\|_\infty \leq \frac{k-\delta}{1-\delta}\epsilon$ where $\boldsymbol{x}^* = \frac{1}{k}\sum_{i \in C_k} \boldsymbol{e}_i$.*

*Proof.* Since $\hat{\boldsymbol{x}}$ is supported on $C_k$, let $i$ be a coordinate that $\hat{\boldsymbol{x}}^*$ puts the least probability mass on; that is, $i \in \arg\min_{j \in C_k} \hat{x}_j$. Considering the utility of playing action $a_i$, we have

$$u(\boldsymbol{e}_i, \hat{\boldsymbol{x}}) = \hat{x}_i \cdot \delta + (1 - \hat{x}_i) \cdot 1 = 1 - \hat{x}_i + \hat{x}_i \delta.$$

Moreover, let $j \in C_k$ be a coordinate such that $\hat{x}_j \geq \frac{1}{k}$. It should hold that

$$u(\boldsymbol{e}_j, \hat{\boldsymbol{x}}) = 1 - (1 - \delta)\hat{x}_j \leq 1 - \frac{1}{k} + \frac{\delta}{k}.$$

Since $(\hat{\boldsymbol{x}}, \hat{\boldsymbol{x}})$ is an $\epsilon$-well-supported NE, we have

$$u(\boldsymbol{e}_j, \hat{\boldsymbol{x}}) \geq u(\boldsymbol{e}_i, \hat{\boldsymbol{x}}) - \epsilon$$
$$\Rightarrow \quad 1 - \frac{1}{k} + \frac{\delta}{k} + \epsilon \geq 1 - \hat{x}_i + \hat{x}_i \delta$$
$$\Rightarrow \quad \hat{x}_i \geq \frac{1}{k} - \frac{\epsilon}{1 - \delta}.$$

Thus, for all coordinates $i \in C_k$, we have $\frac{1}{k} - \frac{\epsilon}{1-\delta} \leq \hat{x}_i \leq \frac{1}{k} + \frac{\epsilon}{1-\delta} \cdot (k-1)$. It then holds that

$$u(\hat{\boldsymbol{x}}, \hat{\boldsymbol{x}}) \geq u(\boldsymbol{e}_i, \hat{\boldsymbol{x}}) - \epsilon$$
$$\geq \left(\frac{1}{k} + \frac{\epsilon}{1-\delta} \cdot (k-1)\right) \cdot \delta + \left(1 - \frac{1}{k} - \frac{\epsilon}{1-\delta} \cdot (k-1)\right) - \epsilon$$
$$\geq 1 - \frac{1}{k} + \frac{\delta}{k} - \left(\frac{k-\delta}{1-\delta}\right)\epsilon.$$

$\square$

**Assumption A.13.** *For the rest of this subsection, we set the parameters as follows:*

- $n \geq k \geq 10$;
- $\epsilon < \delta(1-\delta)/6n^7$;
- $\delta := 1/2$.

(Using the symbolic value of $\delta$ is more convenient in our derivations below.)

*Proof of Lemma A.11.* We let $\boldsymbol{x}^* = \frac{1}{k}\sum_{i \in C_k} \boldsymbol{e}_i$. The proof considers the following three cases:

- The symmetric $\epsilon$-well-supported NE $(\hat{\boldsymbol{x}}, \hat{\boldsymbol{x}})$ has support size less than $k$.
- The symmetric $\epsilon$-well-supported NE $(\hat{\boldsymbol{x}}, \hat{\boldsymbol{x}})$ has support size greater than $k$.
- The symmetric $\epsilon$-well-supported NE $(\hat{\boldsymbol{x}}, \hat{\boldsymbol{x}})$ has support size equal to $k$ but is not supported on a clique.

We proceed to show that for any of the three cases above, we would not be able to have a symmetric $\epsilon$-well-supported NE that achieves value greater than $1 - \frac{1}{k} + \frac{\delta}{k} - \frac{2\delta}{n^2 k^4} + 2\epsilon$, which is a contradiction.

- For the first case, since the support has size less than $k$, we can find a coordinate $i \in [n]$ such that $\hat{x}_i \geq \frac{1}{k-1}$. Therefore, the value of playing that action is

$$u(\boldsymbol{e}_i, \hat{\boldsymbol{x}}) \leq 1 - \frac{1}{k-1} + \frac{\delta}{k-1}.$$

Since $(\hat{\boldsymbol{x}}, \hat{\boldsymbol{x}})$ is a symmetric $\epsilon$-well-supported NE, we have

$$u(\hat{\boldsymbol{x}}, \hat{\boldsymbol{x}}) \leq u(\boldsymbol{e}_i, \hat{\boldsymbol{x}}) + \epsilon$$
$$\leq 1 - \frac{1}{k-1} + \frac{\delta}{k-1} + \epsilon$$
$$\leq 1 - \frac{1}{k} + \frac{\delta}{k} - \frac{2\delta}{n^2 k^4} + 2\epsilon,$$

where in the last step we use Assumption A.13.

- For the second case, suppose $|\mathrm{supp}(\hat{\boldsymbol{x}})| = m > k$. By Lemma A.16, since the maximum clique size is $k$, we can find a set $\mathcal{S} \subseteq \mathrm{supp}(\hat{\boldsymbol{x}})$ with at least $m - k + 1$ elements such that for each coordinate $i \in \mathcal{S}$, we can find a coordinate $j \in \mathcal{S}$ such that $\overline{\mathbf{A}}_{i,j} = \overline{\mathbf{A}}_{j,i} = 0$. Now we consider the utility of playing action $a_i$ and $a_j$, we have

$$u(\boldsymbol{e}_i, \hat{\boldsymbol{x}}) = \sum_{l \in \mathrm{supp}(\hat{\boldsymbol{x}}) - \{i,j\}} \hat{x}_l \cdot \overline{\mathbf{A}}_{i,l} + \hat{x}_i \cdot \delta + \hat{x}_j \cdot \overline{\mathbf{A}}_{i,j},$$
$$u(\boldsymbol{e}_j, \hat{\boldsymbol{x}}) = \sum_{l \in \mathrm{supp}(\hat{\boldsymbol{x}}) - \{i,j\}} \hat{x}_l \cdot \overline{\mathbf{A}}_{j,l} + \hat{x}_j \cdot \delta + \hat{x}_i \cdot \overline{\mathbf{A}}_{i,j}.$$

Since $(\hat{\boldsymbol{x}}, \hat{\boldsymbol{x}})$ is a symmetric $\epsilon$-well-supported NE, we have

$$\sum_{l \in \mathrm{supp}(\hat{\boldsymbol{x}}) - \{i,j\}} \hat{x}_l \cdot \overline{\mathbf{A}}_{i,l} + \hat{x}_i \cdot \delta + \hat{x}_j \cdot \overline{\mathbf{A}}_{i,j} \geq \sum_{l \in \mathrm{supp}(\hat{\boldsymbol{x}}) - \{i,j\}} \hat{x}_l \cdot \overline{\mathbf{A}}_{j,l} + \hat{x}_j \cdot \delta + \hat{x}_i \cdot \overline{\mathbf{A}}_{i,j} - \epsilon$$
$$\Rightarrow \sum_{l \in \mathrm{supp}(\hat{\boldsymbol{x}}) - \{i,j\}} \hat{x}_l \cdot \overline{\mathbf{A}}_{i,l} - \sum_{l \in \mathrm{supp}(\hat{\boldsymbol{x}}) - \{i,j\}} \hat{x}_l \cdot \overline{\mathbf{A}}_{j,l} \geq \hat{x}_j \cdot \delta - \hat{x}_i \cdot \delta - \epsilon.$$

Now, by moving all the probability mass from action $j$ to action $i$, we form a new strategy $\boldsymbol{x}' = \hat{\boldsymbol{x}} + \hat{x}_j \cdot \boldsymbol{e}_i - \hat{x}_j \cdot \boldsymbol{e}_j$ such that

$$u(\boldsymbol{x}', \boldsymbol{x}') - u(\hat{\boldsymbol{x}}, \hat{\boldsymbol{x}}) = 2\hat{x}_j \left( \sum_{l \in \mathrm{supp}(\hat{\boldsymbol{x}}) - \{i,j\}} \hat{x}_l \cdot \overline{\mathbf{A}}_{i,l} - \sum_{l \in \mathrm{supp}(\hat{\boldsymbol{x}}) - \{i,j\}} \hat{x}_l \cdot \overline{\mathbf{A}}_{j,l} \right) + 2\hat{x}_i \cdot \hat{x}_j \cdot \delta$$
$$\geq 2\hat{x}_j \cdot ((\hat{x}_j - \hat{x}_i)\delta - \epsilon) + 2\hat{x}_i \cdot \hat{x}_j \cdot \delta$$
$$\geq 2\hat{x}_j^2 \delta - 2\epsilon. \tag{30}$$

Suppose there is a coordinate $i \in \mathcal{S}$ such that $\hat{x}_i \geq \frac{1}{nk^2}$; then, from (30), we have

$$u(\hat{\boldsymbol{x}}, \hat{\boldsymbol{x}}) \leq u(\boldsymbol{x}^*, \boldsymbol{x}^*) - 2 \cdot (\frac{1}{nk^2})^2 \cdot \delta + 2\epsilon$$
$$= 1 - \frac{1}{k} + \frac{\delta}{k} - \frac{2\delta}{n^2 k^4} + 2\epsilon.$$

If $\hat{x}_i < \frac{1}{nk^2}$ for any $i \in \mathcal{S}$, then there exists a coordinate $l \notin \mathcal{S}$ such that $\hat{x}_l > \frac{1 - (m-k+1) \cdot \frac{1}{nk^2}}{m - (m-k+1)} \geq \frac{1}{k} + \frac{1}{k^2}$. Then, considering the utility when playing action $a_l$,

$$u(\boldsymbol{e}_l, \hat{\boldsymbol{x}}) \leq 1 \cdot (1 - \frac{1}{k} - \frac{1}{k^2}) + \delta \cdot (\frac{1}{k} + \frac{1}{k^2})$$
$$= 1 - \frac{1}{k} + \frac{\delta}{k} - \frac{1}{k^2} + \frac{\delta}{k^2}$$
$$< 1 - \frac{1}{k} + \frac{\delta}{k} - \frac{2\delta}{n^2 k^4}, \tag{31}$$

where in (31) we used Assumption A.13.

Since the $l$th action is played with positive probability and $(\hat{\boldsymbol{x}}, \hat{\boldsymbol{x}})$ is an $\epsilon$-well-supported NE, we have $u(\hat{\boldsymbol{x}}, \hat{\boldsymbol{x}}) \leq u(\boldsymbol{e}_l, \hat{\boldsymbol{x}}) + \epsilon < 1 - \frac{1}{k} + \frac{\delta}{k} - \frac{2\delta}{n^2 k^4} + 2\epsilon.$

- For the third case, since the support is not on a clique, the exists at least coordinates $i, j$ such that $\hat{x}_i > 0, \hat{x}_j > 0$, and $\overline{\mathbf{A}}_{i,j} = \overline{\mathbf{A}}_{j,i} = 0$. Similarly as case two, if $\hat{x}_i \geq \frac{1}{nk^2}$ or $\hat{x}_j \geq \frac{1}{nk^2}$, then we have

$$u(\hat{\boldsymbol{x}}, \hat{\boldsymbol{x}}) \leq u(\boldsymbol{x}^*, \boldsymbol{x}^*) - 2 \cdot \left(\frac{1}{nk^2}\right)^2 \cdot \delta + 2\epsilon$$

$$= 1 - \frac{1}{k} + \frac{\delta}{k} - \frac{2\delta}{n^2 k^4} + 2\epsilon.$$

If $\hat{x}_i < \frac{1}{nk^2}$ and $\hat{x}_j < \frac{1}{nk^2}$, then there exists an coordinate $l$ such that $\hat{x}_l \geq \frac{1 - 2 \cdot \frac{1}{nk^2}}{k-2} > \frac{1}{k} + \frac{1}{k^2}$. Same as (31), we conclude that $u(\hat{\boldsymbol{x}}, \hat{\boldsymbol{x}})$ is at most $1 - \frac{1}{k} + \frac{\delta}{k} - \frac{2\delta}{n^2 k^4} + 2\epsilon$.

The proof is complete. $\qquad\square$

We now construct a new symmetric identical payoff game $(\mathbf{B}, \mathbf{B})$, where $\mathbf{B}$ is defined as

$$\mathbf{B} = \begin{bmatrix} \overline{\mathbf{A}}_{1,1} & \cdots & \overline{\mathbf{A}}_{1,n} & r \\ \vdots & \ddots & \vdots & \vdots \\ \overline{\mathbf{A}}_{n,1} & \cdots & \overline{\mathbf{A}}_{n,n} & r \\ r & \cdots & r & V \end{bmatrix}. \tag{32}$$

Above, $V := 1 - \frac{1}{k} + \frac{\delta}{k}$ and $r := 1 - \frac{1}{k} + \frac{\delta}{k} - \frac{\delta}{n^2 k^4} + 3\epsilon$. Similarly to our discussion after Theorem A.8, it follows that the symmetric (exact) Nash equilibria of this game can only be in one of the following forms:

1. $(\boldsymbol{x}^*, \boldsymbol{x}^*)$ with $\boldsymbol{x}^* := \boldsymbol{e}_{n+1}$;
2. $(\boldsymbol{x}^*, \boldsymbol{x}^*)$ with $\boldsymbol{x}^* := \frac{1}{k} \sum_{i \in C_k} \boldsymbol{e}_i$, where $C_k \subseteq [n]$ is a clique in $G$ of size $k$ ;
3. $(\boldsymbol{x}^*, \boldsymbol{x}^*)$ with $\boldsymbol{x}^* := \frac{1}{2} \boldsymbol{e}_{n+1} + \frac{1}{2k} \sum_{i \in C_k} \boldsymbol{e}_i$, where $C_k \subseteq [n]$ is a clique in $G$ of size $k$ .

We now show the following lemma.

**Lemma A.14.** *For any $\epsilon$-well-supported NE $(\hat{\boldsymbol{x}}, \hat{\boldsymbol{x}})$ in game $(\mathbf{B}, \mathbf{B})$, it holds that $\|\hat{\boldsymbol{x}} - \boldsymbol{x}^*\|_\infty \leq 2n^6 \epsilon$, where $(\boldsymbol{x}^*, \boldsymbol{x}^*)$ is an exact NE in one of the three cases above (1,2,3).*

*Proof.* First, we observe that since $V > r$, clearly $(\boldsymbol{e}_{n+1}, \boldsymbol{e}_{n+1})$ is a $\epsilon$-well-supported NE; in this case, $\|\boldsymbol{x}' - \boldsymbol{x}^*\|_\infty = 0$. Furthermore, since $r = 1 - \frac{1}{k} + \frac{\delta}{k} - \frac{\delta}{n^2 k^4} + 3\epsilon$, the game does not attain any $\epsilon$-NE with value less than $1 - \frac{1}{k} + \frac{\delta}{k} - \frac{\delta}{n^2 k^4} + 2\epsilon$. Suppose the game admits an symmetric $\epsilon$-well-supported NE $(\hat{\boldsymbol{x}}, \hat{\boldsymbol{x}})$ where $\hat{\boldsymbol{x}}$ is supported only on the first $n$ actions. Since $u(\hat{\boldsymbol{x}}, \hat{\boldsymbol{x}}) \geq 1 - \frac{1}{k} + \frac{\delta}{k} - \frac{\delta}{n^2 k^4} + 2\epsilon > 1 - \frac{1}{k} + \frac{\delta}{k} - \frac{2\delta}{n^2 k^4} + 2\epsilon$, taking $\boldsymbol{x}^*$ as in (2), we conclude that $\|\hat{\boldsymbol{x}} - \boldsymbol{x}^*\|_\infty \leq \frac{k-1}{1-\delta}\epsilon < 2n^6 \epsilon$ from Lemma A.11.

We proceed to the case where there is a mixed symmetric $\epsilon$-well-supported Nash $(\hat{\boldsymbol{x}}, \hat{\boldsymbol{x}})$ between the last action and the rest of actions such that $0 < \hat{x}_{n+1} < 1$. Denote $\hat{x}_{n+1} = \alpha$ and $\eta(\cdot)$ to denote the renormalization operation. Since $\left(\eta(\hat{\boldsymbol{x}}_{[1\cdots n]}), \eta(\hat{\boldsymbol{x}}_{[1\cdots n]})\right)$ is a symmetric strategy profile, from McLennan and Tourky [2010, Proposition 4], we conclude that there is at least one coordinate $i \in \text{supp}(\hat{\boldsymbol{x}}) - \{n+1\}$ such that

$$u_A\left(\boldsymbol{e}_i, \eta(\hat{\boldsymbol{x}}_{[1\cdots n]})\right) \leq 1 - \frac{1}{k} + \frac{\delta}{k} = V,$$

where $u_A$ is the utility when the payoff matrix is $\overline{\mathbf{A}}$. Since $\left(\eta(\hat{\boldsymbol{x}}_{[1\cdots n]}), \eta(\hat{\boldsymbol{x}}_{[1\cdots n]})\right)$ is an $\epsilon$-well-supported NE, we have

$$u_B(\boldsymbol{e}_i, \hat{\boldsymbol{x}}) \geq u_B(\boldsymbol{e}_{n+1}, \hat{\boldsymbol{x}}) - \epsilon$$
$$\Rightarrow \quad (1-\alpha) \cdot V + \alpha \cdot r \geq (1-\alpha)r + \alpha V - \epsilon$$
$$\Rightarrow \quad \alpha \leq \frac{1}{2} + \frac{\epsilon}{2(V-r)}, \tag{33}$$

where $u_B$ is the utility function when the payoff matrix is $\mathbf{B}$. Plugging in the value of $V$ and $r$ and using Assumption A.13, we find that $\alpha \leq \frac{1}{2} + 2n^6 \epsilon \leq \frac{2}{3}$.

Now, observe that for any action in the support other than the last action $a_i$, the utility of playing such action $u_B(a_i, \hat{\boldsymbol{x}}) = u_A(a_i, \hat{\boldsymbol{x}}_{[1\cdots n]}) + r\alpha$. Since $(\hat{\boldsymbol{x}}, \hat{\boldsymbol{x}})$ is an $\epsilon$-well-supported Nash Equilibrium, we have $u_B(a_i, \hat{\boldsymbol{x}}) \geq \max_j u_B(a_j, \hat{\boldsymbol{x}}) - \epsilon$ for all pairs $(i, j) \in \text{supp}(\hat{\boldsymbol{x}})$. Since $\alpha \leq \frac{2}{3}$, it follows that $u_A\left(a_i, \eta(\hat{\boldsymbol{x}}_{[1\cdots n]})\right) \geq \max_j u_A\left(a_j, \eta(\hat{\boldsymbol{x}}_{[1\cdots n]})\right) - 3\epsilon$ for any pairs $(i, j) \in \text{supp}(\hat{\boldsymbol{x}}) - \{n+1\}$. Thus, we conclude that $\left(\eta(\hat{\boldsymbol{x}}_{[1\cdots n]}), \eta(\hat{\boldsymbol{x}}_{[1\cdots n]})\right)$ forms a symmetric $3\epsilon$-well-supported Nash Equilibrium in game $(\overline{\mathbf{A}}, \overline{\mathbf{A}})$. Further, the value of playing the last action is $(1 - \alpha)r + \alpha V > r$, and so the only situation where there is a mixed Nash between the last action and the rest actions is when $u_A\left((\eta(\hat{\boldsymbol{x}}_{[1\cdots n]}), \eta(\hat{\boldsymbol{x}}_{[1\cdots n]}))\right) \geq r - \epsilon$. Therefore, by Lemma A.12 and Lemma A.11, we conclude that $\hat{\boldsymbol{x}}_{[1\cdots n]}$ is supported on a clique of size $k$. There exits at least one coordinate $i \in [n]$, with $0 < \hat{x}_i$ and $\hat{x}_i \geq \frac{1}{k}$, such that

$$u_A(\boldsymbol{e}_i, \eta(\hat{\boldsymbol{x}}_{[1\cdots n]})) \geq 1 - \frac{1}{k} + \frac{\delta}{k} = V.$$

Since $(\hat{\boldsymbol{x}}, \hat{\boldsymbol{x}})$ is an $\epsilon$-well-supported Nash, we have $u(\boldsymbol{e}_i, \hat{\boldsymbol{x}}) \leq u(\boldsymbol{e}_{n+1}, \hat{\boldsymbol{x}}) + \epsilon$, and so this gives

$$(1 - \alpha) \cdot V + \alpha \cdot r \leq (1 - \alpha) \cdot r + \alpha \cdot V + \epsilon \tag{34}$$

$$\Rightarrow \quad \alpha \geq \frac{1}{2} - \frac{\epsilon}{2(V - r)}. \tag{35}$$

Using Assumption A.13 and combining with (33),

$$\frac{1}{2} - 2n^6\epsilon \leq \alpha \leq \frac{1}{2} + 2n^6\epsilon.$$

By taking $\boldsymbol{x}^*$ as in (3), we conclude that $\|\hat{\boldsymbol{x}} - \boldsymbol{x}^*\|_\infty \leq 2n^6\epsilon$. $\qquad \square$

**Theorem A.15.** *For any $\epsilon$-NE $(\boldsymbol{x}, \boldsymbol{x})$ in game $(\mathbf{B}, \mathbf{B})$, it holds that $\|\boldsymbol{x} - \boldsymbol{x}^*\|_\infty \leq n^6\sqrt{\epsilon}$, where $(\boldsymbol{x}^*, \boldsymbol{x}^*)$ is an exact NE in one of the forms specified above (1,2,3).*

*Proof.* Chen et al. [2009, Lemma 3.2] showed that from any $\epsilon^2/8$-NE $(\boldsymbol{x}, \boldsymbol{y})$ in any two player bimatrix game, one can construct (in polynomial time) an $\epsilon$-well-supported NE $(\boldsymbol{x}', \boldsymbol{y}')$ such that $\|\boldsymbol{x} - \boldsymbol{x}'\|_\infty \leq \frac{\epsilon}{4}$ and $\|\boldsymbol{y} - \boldsymbol{y}'\|_\infty \leq \frac{\epsilon}{4}$. Setting $\epsilon' := \frac{\epsilon^2}{8}$ for the $\epsilon$ defined in Lemma A.14, the proof follows. $\qquad \square$

The proof of Theorem 4.7 follows directly by observing that having two symmetric $\epsilon$-NE $(\boldsymbol{x}, \boldsymbol{x})$ and $(\boldsymbol{y}, \boldsymbol{y})$ such that $\|\boldsymbol{x} - \boldsymbol{y}\|_\infty > 2n^6\sqrt{\epsilon}$ would imply that the game $(\mathbf{B}, \mathbf{B})$ has two distinct exact NE, which in turn implies that there is a clique of size $k$ in the graph.

We next state and prove an auxiliary lemma that was used earlier.

**Lemma A.16.** *For any graph $G = (V, E)$ with $n$ vertices, if the maximum clique has size $k$, then we can form a set $\mathcal{S} \subseteq V$ of size at least $n - k + 1$ such that for any vertex $i \in \mathcal{S}$, there exists a vertex $j \in \mathcal{S}$ such that $i$ and $j$ are not connected.*

*Proof.* Suppose the largest set $\mathcal{S}$ we can form has cardinality $|\mathcal{S}| < n - k + 1$, this implies there is a set $\mathcal{S}' = V - \mathcal{S}$ with at least $n - (n - k) = k$ vertices such that each vertex in $\mathcal{S}'$ is connected to all other vertices in $G$. However if this is the case, there is at least one vertex $v \notin \mathcal{S}'$ that are connected to all vertices in $\mathcal{S}'$. This contradicts the fact that maximum clique has size $k$. $\qquad \square$

We proceed now with the proof of Theorem 1.6.

*Proof of Theorem 1.6.* It suffices to consider the antisymmetric function $f(\boldsymbol{x}, \boldsymbol{y}) := \boldsymbol{y}^\top \mathbf{B}\boldsymbol{y} - \boldsymbol{x}^\top \mathbf{B}\boldsymbol{x}$, where symmetric matrix $\mathbf{B}$ is defined as in (29), using our new matrix $\overline{\mathbf{A}}$ instead of $\mathbf{A}$ (see (32)). Any $\epsilon$-first-order Nash equilibrium $(\boldsymbol{x}^*, \boldsymbol{y}^*)$ of this (separable) min-max optimization problem induces, two symmetric $\epsilon$-Nash equilibria—namely, $(\boldsymbol{x}^*, \boldsymbol{x}^*)$ and $(\boldsymbol{y}^*, \boldsymbol{y}^*)$—in the symmetric, identical-interest, game $(\mathbf{B}, \mathbf{B})$. Using Theorem 4.7, the claim follows. $\qquad \square$

## A.5 Proofs from Section 4.3

We conclude with the missing proofs from Section 4.3. We first introduce two lemmas which are useful in later proofs.

The first key lemma, which mirrors Lemma 3.2, shows that, in an approximate Nash equilibrium of (4), $\boldsymbol{x} \approx \boldsymbol{y}$ and $\hat{\boldsymbol{x}} \approx \hat{\boldsymbol{y}}$. This is crucial as it allows us to construct—up to some small error—quadratic terms in the utility function, as in our hardness result for symmetric min-max optimization.

**Lemma A.17.** *Let* $(\boldsymbol{x}^*, \boldsymbol{y}^*, \boldsymbol{z}^*, \hat{\boldsymbol{x}}^*, \hat{\boldsymbol{y}}^*, \hat{\boldsymbol{z}}^*)$ *be an* $\epsilon^2$*-Nash equilibrium of* (4) *with* $\epsilon^2 \le 1/2$*. Then,* $\|\boldsymbol{x}^* - \boldsymbol{y}^*\|_\infty \le 2\epsilon$ *and* $\|\hat{\boldsymbol{x}}^* - \hat{\boldsymbol{y}}^*\|_\infty \le 2\epsilon$*.*

*Proof.* For the sake of contradiction, suppose that $\|\boldsymbol{x}^* - \boldsymbol{y}^*\|_\infty > 2\epsilon$. Without loss of generality, let us further assume that there is some coordinate $i$ such that $x_i^* - y_i^* > 2\epsilon$. The payoff difference for Player $\hat{\boldsymbol{z}}$ when playing the $i$th action compared to action $a_{2n+1}$ reads

$$u(\boldsymbol{x}^*, \boldsymbol{y}^*, \boldsymbol{z}^*, \hat{\boldsymbol{x}}^*, \hat{\boldsymbol{y}}^*, a_i) - u(\boldsymbol{x}^*, \boldsymbol{y}^*, \boldsymbol{z}^*, \hat{\boldsymbol{x}}^*, \hat{\boldsymbol{y}}^*, a_{2n+1}) = \frac{|\mathbf{A}_{\min}|}{\epsilon} \cdot (x_i^* - y_i^*) - |\mathbf{A}_{\min}|$$
$$> |\mathbf{A}_{\min}|.$$

By Lemma A.1, it follows that $\hat{z}_{2n+1}^* < \frac{\epsilon^2}{|\mathbf{A}_{\min}|} \le \epsilon^2$. Moreover, given that $(\boldsymbol{x}^*, \boldsymbol{y}^*, \boldsymbol{z}^*, \hat{\boldsymbol{x}}^*, \hat{\boldsymbol{y}}^*, \hat{\boldsymbol{z}}^*)$ is an $\epsilon^2$-Nash equilibrium, we have

$$
\begin{aligned}
u(\boldsymbol{x}^*, \boldsymbol{y}^*, \boldsymbol{z}^*, \hat{\boldsymbol{x}}^*, \hat{\boldsymbol{y}}^*, \hat{\boldsymbol{z}}^*) &\ge u(\boldsymbol{x}^*, \boldsymbol{y}^*, \boldsymbol{z}^*, \hat{\boldsymbol{x}}^*, \hat{\boldsymbol{y}}^*, a_i) - \epsilon^2 \\
&= \langle \boldsymbol{x}^*, \mathbf{A}\boldsymbol{y}^* \rangle - \langle \hat{\boldsymbol{x}}^*, \mathbf{A}\hat{\boldsymbol{y}}^* \rangle + \langle \boldsymbol{x}^*, \mathbf{C}\hat{\boldsymbol{x}}^* \rangle \\
&\quad + \frac{|\mathbf{A}_{\min}|}{\epsilon}(x_i^* - y_i^*) - \delta(\hat{\boldsymbol{x}}, \hat{\boldsymbol{y}}, \boldsymbol{z}) - \epsilon^2 \\
&\ge \mathbf{A}_{\min} - \langle \hat{\boldsymbol{x}}^*, \mathbf{A}\hat{\boldsymbol{y}}^* \rangle + \langle \boldsymbol{x}^*, \mathbf{C}\hat{\boldsymbol{x}}^* \rangle + 2|\mathbf{A}_{\min}| - \delta(\hat{\boldsymbol{x}}, \hat{\boldsymbol{y}}, \boldsymbol{z}) - \epsilon^2 \\
&= |\mathbf{A}_{\min}| - \langle \hat{\boldsymbol{x}}^*, \mathbf{A}\hat{\boldsymbol{y}}^* \rangle + \langle \boldsymbol{x}^*, \mathbf{C}\hat{\boldsymbol{x}}^* \rangle - \delta(\hat{\boldsymbol{x}}, \hat{\boldsymbol{y}}, \boldsymbol{z}) - \epsilon^2.
\end{aligned}
$$

Now, considering the deviation of Player $\boldsymbol{y}$ to $\boldsymbol{y}' := \boldsymbol{x}^*$,

$$
\begin{aligned}
u(\boldsymbol{x}^*, \boldsymbol{y}', \boldsymbol{z}^*, \hat{\boldsymbol{x}}^*, \hat{\boldsymbol{y}}^*, \hat{\boldsymbol{z}}^*) - u(\boldsymbol{x}^*, \boldsymbol{y}^*, \boldsymbol{z}^*, \hat{\boldsymbol{x}}^*, \hat{\boldsymbol{y}}^*, \hat{\boldsymbol{z}}^*) &\le \langle \boldsymbol{x}^*, \mathbf{A}\boldsymbol{x}^* \rangle + \epsilon^2 |\mathbf{A}_{\min}| - |\mathbf{A}_{\min}| + \epsilon^2 \\
&\le -2 + 2\epsilon^2 \\
&< -\epsilon^2,
\end{aligned}
$$

which contradicts the fact that $(\boldsymbol{x}^*, \boldsymbol{y}^*, \boldsymbol{z}^*, \hat{\boldsymbol{x}}^*, \hat{\boldsymbol{y}}^*, \hat{\boldsymbol{z}}^*)$ is an $\epsilon^2$-Nash equilibrium. We conclude that $\|\boldsymbol{x}^* - \boldsymbol{y}^*\|_\infty \le 2\epsilon$; the proof for the fact that $\|\hat{\boldsymbol{x}}^* - \hat{\boldsymbol{y}}^*\|_\infty \le 2\epsilon$ follows similarly. $\qquad\square$

Next, following the argument of Lemma 3.3, we show that, in equilibrium, Players $\boldsymbol{z}$ and $\hat{\boldsymbol{z}}$ place most of their probability mass on action $a_{2n+1}$, thereby having only a small effect on the game between Players $\boldsymbol{x}$ and $\boldsymbol{y}$ vs. $\hat{\boldsymbol{x}}$ and $\hat{\boldsymbol{y}}$.

**Lemma A.18.** *Let* $(\boldsymbol{x}^*, \boldsymbol{y}^*, \boldsymbol{z}^*, \hat{\boldsymbol{x}}^*, \hat{\boldsymbol{y}}^*, \hat{\boldsymbol{z}}^*)$ *be an* $\epsilon^2$*-Nash equilibrium of* (4) *with* $\epsilon \le 1/10$*. Then,* $z_j, \hat{z}_j \le 9\epsilon$ *for all* $j \in [2n]$*.*

*Proof.* We will prove that $z_j \le 9\epsilon$ for all $j \in [2n]$; the corresponding claim for Player $\hat{\boldsymbol{z}}$ follows similarly. Fix $i \in [n]$. Lemma A.17 shows that $|y_i^* - x_i^*| \le 2\epsilon$. We shall consider two cases.

First, suppose that $|y_i^* - x_i^*| \le \epsilon/2$. Then,

$$
\begin{aligned}
u(\boldsymbol{x}^*, \boldsymbol{y}^*, \boldsymbol{z}^*, \hat{\boldsymbol{x}}^*, \hat{\boldsymbol{y}}^*, a_{2n+1}) - u(\boldsymbol{x}^*, \boldsymbol{y}^*, \boldsymbol{z}^*, \hat{\boldsymbol{x}}^*, \hat{\boldsymbol{y}}^*, a_i) &\ge |\mathbf{A}_{\min}| - \frac{|\mathbf{A}_{\min}|}{\epsilon} \cdot (x_i^* - y_i^*) \\
&\ge \frac{|\mathbf{A}_{\min}|}{2} \ge \frac{1}{2}.
\end{aligned}
$$

By Lemma A.1, it follows that $\hat{z}_i \le 2\epsilon^2$, and similar reasoning yields $\hat{z}_{n+i} \le 2\epsilon^2$. On the other hand, suppose that $|y_i^* - x_i^*| > \epsilon/2$. Without loss of generality, we can assume that $y_i^* - x_i^* \ge 0$; the contrary case is symmetric. Since $\boldsymbol{x}^* \in \Delta^n$ and $\boldsymbol{y}^* \in \Delta^n$, there is some coordinate $j \in [n]$ such that

$y_j^* - x_j^* < 0$. As before, by Lemma A.1, it follows that $\hat{z}_i^* \leq \epsilon^2$ and $\hat{z}_{n+j}^* \leq \epsilon^2$. Now, we consider the deviation

$$\Delta^n \ni \boldsymbol{y}' = \boldsymbol{y}^* + (x_i^* - y_i^*)\boldsymbol{e}_i + (y_i^* - x_i^*)\boldsymbol{e}_j.$$

Then, we have

$$u(\boldsymbol{x}^*, \boldsymbol{y}', \boldsymbol{z}^*, \hat{\boldsymbol{x}}^*, \hat{\boldsymbol{y}}^*, \hat{\boldsymbol{z}}^*) - u(\boldsymbol{x}^*, \boldsymbol{y}^*, \boldsymbol{z}^*, \hat{\boldsymbol{x}}^*, \hat{\boldsymbol{y}}^*, \hat{\boldsymbol{z}}^*)$$

$$= \langle \boldsymbol{x}^*, \mathbf{A}(\boldsymbol{y}' - \boldsymbol{y}^*) \rangle + \frac{|\mathbf{A}_{\min}|}{\epsilon} \left( \hat{z}_i^*(x_i^* - y_i' - (x_i^* - y_i^*)) + \hat{z}_j^*(x_j^* - y_j' - (x_j^* - y_j^*)) \right)$$

$$+ \frac{|\mathbf{A}_{\min}|}{\epsilon} \left( \hat{z}_{n+i}^*(y_i' - x_i^* - (y_i^* - x_i^*)) + \hat{z}_{n+j}^*(y_j' - x_j^* - (y_j^* - x_j^*)) \right)$$

$$= \langle \boldsymbol{x}^*, \mathbf{A}(\boldsymbol{y}' - \boldsymbol{y}^*) \rangle + \frac{|\mathbf{A}_{\min}|}{\epsilon} \left( \hat{z}_i^*(y_i^* - y_i') + \hat{z}_j^*(y_j^* - y_j') + \hat{z}_{n+i}^*(y_i' - y_i^*) + \hat{z}_{n+j}^*(y_j' - y_j^*) \right)$$

$$\leq 4\epsilon |\mathbf{A}_{\min}| + \frac{|\mathbf{A}_{\min}|}{\epsilon} \left( \epsilon^2 \cdot 2\epsilon + \hat{z}_{n+i}^* \cdot \left( -\frac{\epsilon}{2} \right) + \epsilon^2 \cdot 2\epsilon \right)$$

$$\leq - \left( \frac{1}{2} \hat{z}_{n+i}^* - 4\epsilon^2 - 4\epsilon \right) |\mathbf{A}_{\min}|.$$

At the same time, since $(\boldsymbol{x}^*, \boldsymbol{y}^*, \boldsymbol{z}^*, \hat{\boldsymbol{x}}^*, \hat{\boldsymbol{y}}^*, \hat{\boldsymbol{z}}^*)$ is an $\epsilon^2$-Nash equilibrium, we have

$$u(\boldsymbol{x}^*, \boldsymbol{y}', \boldsymbol{z}^*, \hat{\boldsymbol{x}}^*, \hat{\boldsymbol{y}}^*, \hat{\boldsymbol{z}}^*) - u(\boldsymbol{x}^*, \boldsymbol{y}^*, \boldsymbol{z}^*, \hat{\boldsymbol{x}}^*, \hat{\boldsymbol{y}}^*, \hat{\boldsymbol{z}}^*) \geq -\epsilon^2.$$

Thus,

$$\left( \frac{1}{2} \hat{z}_{n+i}^* - 4\epsilon^2 - 4\epsilon \right) |\mathbf{A}_{\min}| \leq \epsilon^2 \implies \hat{z}_{n+i}^* \leq 9\epsilon.$$

We conclude that $\hat{z}_i^* \leq \epsilon^2$ and $\hat{z}_{n+i}^* \leq 9\epsilon$. The case where $x_i^* - y_i^* \geq 0$ can be treated similarly. $\square$

Armed with those two basic lemmas, we are ready to complete the proof of Theorem 1.7.

*Proof of Theorem 1.7.* Suppose that $(\boldsymbol{x}^*, \boldsymbol{y}^*, \boldsymbol{z}^*, \hat{\boldsymbol{x}}^*, \hat{\boldsymbol{y}}^*, \hat{\boldsymbol{z}}^*)$ is an $\epsilon^2$-Nash equilibrium. We have that for any $\boldsymbol{y}' \in \Delta^n$,

$$\langle \boldsymbol{x}^*, \mathbf{A}\boldsymbol{y}^* \rangle \leq \langle \boldsymbol{x}^*, \mathbf{A}\boldsymbol{y}' \rangle + \frac{|\mathbf{A}_{\min}|}{\epsilon} \left( \sum_{i=1}^n \hat{z}_i^*(y_i^* - y_i') + \hat{z}_{n+i}^*(y_i' - y_i^*) \right) + \epsilon^2. \tag{36}$$

Moreover, considering a deviation from $\boldsymbol{x}^*$ to $\boldsymbol{y}'$,

$$\langle \boldsymbol{x}^*, \mathbf{A}\boldsymbol{y}^* \rangle + \langle \boldsymbol{x}^*, \mathbf{C}\hat{\boldsymbol{x}}^* \rangle \leq \langle \boldsymbol{y}', \mathbf{A}\boldsymbol{y}^* \rangle + \langle \boldsymbol{y}', \mathbf{C}\hat{\boldsymbol{x}}^* \rangle$$

$$+ \frac{|\mathbf{A}_{\min}|}{\epsilon} \left( \sum_{i=1}^n \hat{z}_i^*(y_i' - x_i^*) + \hat{z}_{n+i}^*(x_i^* - y_i') \right) + \epsilon^2. \tag{37}$$

Summing (36) and (37),

$$2\langle \boldsymbol{x}^*, \mathbf{A}\boldsymbol{y}^* \rangle + \langle \boldsymbol{x}^*, \mathbf{C}\hat{\boldsymbol{x}}^* \rangle \leq \langle \boldsymbol{y}', \mathbf{A}(\boldsymbol{x}^* + \boldsymbol{y}^*) \rangle + \langle \boldsymbol{y}', \mathbf{C}\hat{\boldsymbol{x}}^* \rangle$$

$$+ \frac{|\mathbf{A}_{\min}|}{\epsilon} \left( \sum_{i=1}^n \hat{z}_i^*(y_i^* - x_i^*) + \hat{z}_{n+i}^*(x_i^* - y_i^*) \right) + 2\epsilon^2$$

$$\leq 2\langle \boldsymbol{x}^*, \mathbf{A}\boldsymbol{y}' \rangle + \langle \boldsymbol{y}', \mathbf{C}\hat{\boldsymbol{x}}^* \rangle + 2\epsilon n |\mathbf{A}_{\min}| + \frac{|\mathbf{A}_{\min}|}{\epsilon}(2n \cdot 9\epsilon \cdot 2\epsilon) + 2\epsilon^2$$

$$\leq 2\langle \boldsymbol{x}^*, \mathbf{A}\boldsymbol{y}' \rangle + \langle \boldsymbol{y}', \mathbf{C}\hat{\boldsymbol{x}}^* \rangle + (38n + 2)|\mathbf{A}_{\min}|\epsilon. \tag{38}$$

Moreover,

$$\langle \boldsymbol{x}^*, \mathbf{A}\boldsymbol{x}^* \rangle = \langle \boldsymbol{x}^*, \mathbf{A}\boldsymbol{y}^* \rangle + \langle \boldsymbol{x}^*, \mathbf{A}(\boldsymbol{x}^* - \boldsymbol{y}^*) \rangle \leq \langle \boldsymbol{x}^*, \mathbf{A}\boldsymbol{y}^* \rangle + 2|\mathbf{A}_{\min}|n\epsilon. \tag{39}$$

Combining (38) and (39), we get that for all $\boldsymbol{y}' \in \Delta^n$,

$$\langle \boldsymbol{x}^*, \mathbf{A}\boldsymbol{x}^* \rangle + \frac{1}{2}\langle \boldsymbol{x}^*, \mathbf{C}\hat{\boldsymbol{x}}^* \rangle \leq \langle \boldsymbol{y}', \mathbf{A}\boldsymbol{x}^* \rangle + \frac{1}{2}\langle \boldsymbol{y}', \mathbf{C}\hat{\boldsymbol{x}}^* \rangle + (21n + 1)|\mathbf{A}_{\min}|\epsilon. \tag{40}$$

Similarly, we can show that for all $\hat{\boldsymbol{x}}' \in \Delta^n$,

$$\langle -\hat{\boldsymbol{x}}^*, \mathbf{A}\hat{\boldsymbol{x}}^* \rangle + \frac{1}{2}\langle \boldsymbol{x}^*, \mathbf{C}\hat{\boldsymbol{x}}^* \rangle \geq -\langle \hat{\boldsymbol{x}}', \mathbf{A}\hat{\boldsymbol{x}}^* \rangle + \frac{1}{2}\langle \boldsymbol{x}^*, \mathbf{C}\hat{\boldsymbol{x}}' \rangle - (21n+1)|\mathbf{A}_{\min}|\epsilon. \qquad (41)$$

Taking $\boldsymbol{y}' = \hat{\boldsymbol{x}}'$ in (40) and summing with (41), we get that for all $\hat{\boldsymbol{x}}' \in \Delta^n$,

$$\langle \hat{\boldsymbol{x}}', \mathbf{A}\boldsymbol{x}^* \rangle + \langle \hat{\boldsymbol{x}}', \mathbf{A}\hat{\boldsymbol{x}}^* \rangle + \frac{1}{2}(\langle \hat{\boldsymbol{x}}', \mathbf{C}\hat{\boldsymbol{x}}^* \rangle + \langle \hat{\boldsymbol{x}}', \mathbf{C}\boldsymbol{x}^* \rangle) + (42n+2)|\mathbf{A}_{\min}|\epsilon \geq \langle \boldsymbol{x}^*, \mathbf{A}\boldsymbol{x}^* \rangle + \langle \hat{\boldsymbol{x}}^*, \mathbf{A}\hat{\boldsymbol{x}}^* \rangle,$$

where we used the fact that $\mathbf{C}$ is skew-symmetric. Now, using the fact that the Nash equilibrium is symmetric, so that $\boldsymbol{x}^* = \hat{\boldsymbol{x}}^*$, we have

$$\langle \hat{\boldsymbol{x}}', \mathbf{A}\boldsymbol{x}^* \rangle + \frac{1}{2}\langle \hat{\boldsymbol{x}}', \mathbf{C}\boldsymbol{x}^* \rangle + (21n+1)|\mathbf{A}_{\min}|\epsilon \geq \langle \boldsymbol{x}^*, \mathbf{A}\boldsymbol{x}^* \rangle$$

$$\geq \langle \boldsymbol{x}^*, \mathbf{A}\boldsymbol{x}^* \rangle + \frac{1}{2}\langle \boldsymbol{x}^*, \mathbf{C}\boldsymbol{x}^* \rangle. \qquad (42)$$

Setting $\mathbf{A} := -\frac{1}{2}(\mathbf{R} + \mathbf{R}^\top)$ and $\mathbf{C} := \mathbf{R}^\top - \mathbf{R}$, (42) shows that

$$\langle \hat{\boldsymbol{x}}', \mathbf{R}\boldsymbol{x}^* \rangle \leq \langle \boldsymbol{x}^*, \mathbf{R}\boldsymbol{x}^* \rangle + (21n+1)|\mathbf{A}_{\min}|\epsilon$$

for any $\hat{\boldsymbol{x}}' \in \Delta^n$; *ergo*, $(\boldsymbol{x}^*, \boldsymbol{x}^*)$ is a symmetric $(21n+1)|\mathbf{A}_{\min}|\epsilon$-Nash equilibrium of the symmetric (two-player) game $(\mathbf{R}, \mathbf{R}^\top)$, and the proof follows from Theorem 2.2. $\qquad \square$

