# OpenReview forum: "The Complexity of Symmetric Equilibria in Min-Max Optimization and Team Zero-Sum Games"
_NeurIPS.cc/2025/Conference — NeurIPS 2025 spotlight_

### Official Review · Reviewer_JDAf · 2025-06-30

**Clarity:** 3
**Significance:** 3
**Originality:** 3
**Rating:** 5
**Confidence:** 4

**Summary:**

The paper provides many lower bounds related to the problem of computing equilibria of team zero-sum games and min-max optimization, with a focus on symmetric solutions. The authors show that computing $\epsilon$-Nash equilibria in 3-players adversarial team games is CLS-complete. Moreover, they show that computing symmetric first-order equilibria in symmetric min-max optimization is PPAD-complete and computing symmetric $\epsilon$-Nash in symmetric team zero-sum games is PPAD-complete. As a consequence, they show that symmetric dynamics cannot converge to stationary points. Finally, they show that computing non-symmetric equilibrium in min-max optimization is FNP-hard.

**Questions:**

None.

**Ethical Concerns:**

["NO or VERY MINOR ethics concerns only"]

**Final Justification:**

My positive evaluation of the work is unchanged.

**Limitations:**

Yes.

**Quality:**

3

**Strengths And Weaknesses:**

Strengths:

The paper includes a large set of interesting results. All the results are on very simple instances, e.g., small number of players or quadratic functions, making the results stronger. Moreover, the proofs are simple and easy to follow.

Weaknesses:

- Some results are very similar to or heavily based on previous works and not technically very involved.

- The paper could do a better job highlighting the similarities and differences with previous/concurrent works, especially with the other works in the min-max optimization literature.

For instance, Theorem 4.6 should already follow from the reduction in Daskalakis et al. [2021]. Indeed, if the two dynamics start with $x=y$, the two players observe the same gradient also in their construction. Even if not explicitly stated, their hardness result should hold for symmetric equilibria in unconstrained domains.

The differences and similarities with the works of Bernasconi et al. [2024] are not very clear. At least at a high level the construction is very similar. It seems that the main difference  is in the domain, which requires to reduce from different problems and leads to different approximations.

---

> ### Author Rebuttal · Authors · 2025-07-30
>
> We thank the reviewer for their valuable time and positive comments.
>
> > "Some results are very similar to or heavily based on previous works and not technically very involved."
>
> We consider the relative simplicity of our approach as a strength since it will make it easier for future research to build on our paper. In particular, among others, we are able to recover and strengthen the results of Daskalakis et al. [1] that were especially involved. We believe that our simpler framework will enable building on our approach for this fundamental problem.
>
> > "The paper could do a better job highlighting the similarities and differences with previous/concurrent works, especially with the other works in the min-max optimization literature."
>
> Regarding the differences between our work and Daskalakis et al. [1], we highlight the following key improvements: i) as we pointed out above, our reduction and proof are significantly simpler; ii) our hardness result applies even when the accuracy parameter is constant, which is relevant in modern ML applications where one is often content with a crude solution quality; and iii) our class of functions comprises only simple quadratic functions, whereas the class of functions constructed by Daskalakis et al. is very involved.
>
> Also, we point out that the equilibrium notion we consider is stronger. The main issue is that in their setting, when a symmetry constraint $x \approx y$ is enforced, Player $x$ can only unilaterally deviate to a small set of strategies nearby $y$ in order to satisfy the constraint. On the other hand, our notion allows Player $x$ to unilaterally deviate to any other strategy. Essentially, our reduction captures what Daskalakis et al. [1] refer to as a “safe” version of GDA fixed points, which can be different from usual GDA fixed points — Bernasconi et al. [2] provide an explicit separation in Appendix B.1 of their paper. For this reason, we do not believe that our Theorem 4.6 follows directly from the reduction of Daskalakis et al.
>
> Regarding the differences with Bernasconi et al. [2], the starting point of our reduction and the resulting hard class of min-max optimization problems is different from that paper, both in terms of the domain and the underlying objective function. It is important to note that the domain crucially affects the complexity of the problem. For box constraints, solving linear VIs with even a constant accuracy is PPAD-hard [3]. On the other hand, for constant accuracy and under simplex constraints, there is a well-known quasipolynomial-time algorithm [4]; one needs to have inverse polynomial accuracy to get PPAD-hardness under simplex constraints [5]. Besides the difference in the domain, our class of problems is symmetric, which enables us to prove hardness for computing symmetric equilibria.
>
> We will highlight at greater length those differences in the revised version.
>
> ________
>
> References:
>
> [1] Constantinos Daskalakis, Stratis Skoulakis, and Manolis Zampetakis. The complexity of constrained min-max optimization. In Symposium on Theory of Computing (STOC), 2021.
>
> [2] Martino Bernasconi, Matteo Castiglioni, Andrea Celli, and Gabriele Farina. On the role of constraints in the complexity of min-max optimization, 2024.
>
> [3] Aviad Rubinstein. Inapproximability of Nash equilibrium. In Proceedings of the fortyseventh annual ACM symposium on Theory of computing, pp. 409–418, 2015.
>
> [4] Richard J Lipton, Evangelos Markakis, and Aranyak Mehta. Playing large games using simple strategies. In Proceedings of the 4th ACM conference on Electronic commerce, pages 36–41, 2003.
>
> [5] Xi Chen, Xiaotie Deng, and Shang-Hua Teng. Settling the complexity of computing two-player Nash equilibria. J. ACM, 56(3):14:1–14:57, 2009.

---

> > ### Comment · Reviewer_JDAf · 2025-08-05
> >
> > Thanks for the response. I will keep my positive evaluation of the work.

---

### Official Review · Reviewer_1qji · 2025-07-01

**Clarity:** 2
**Significance:** 3
**Originality:** 3
**Rating:** 5
**Confidence:** 3

**Summary:**

The authors considered the hardness of computing symmetric equilibria in zero-sum games and utilized the hardness result in this area to show the hardness of computing equilibria in 3-player (2 versus. 1) adversarial team games. Using a carefully-designed adversarial team game that enforces the strategy of the two players in the same team to be approximately the same, the authors showed that 3-player adversarial games, and in particular 3-player adversarial polymatrix games, can be reduced from calculating a symmetric equilibrium in a symmetric two-player games, which is known to be CLS-complete. The authors then considered the problem of computing Nash equilibria in symmetric n-dimensional min-max optimizations, and showed that the problem is PPAD-complete, by providing a polynomial-time reduction from calculating symmetric Nash Equilibrium in two-player symmetric games, which is known-to be PPAD- complete. Based on this, the authors constructed a 6-player team (3 versus 3) zero-sum polymatrix game, and provided a polynomial-time reduction to computing Nash equilibria in symmetric min-max optimizations, therefore demonstrating that computing Nash equilibrium in this 6-player team zero-sum game is PPAD-complete. Finally, the authors demonstrated that for certain games, it is NP-hard to determine whether or not there exists a approximately symmetric $n^{-c}$-approximate Nash equilibrium.

**Questions:**

* It is interesting that for symmetric games, all accurate Nash equilibria must be symmetric, whereas some approximate Nash equilibria are allowed to be slightly asymmetric. If some regulations are enforced on a symmetric game (say Lipschitz continuity of the payoff), can one understand a bit more about the hardness landscape?

**Ethical Concerns:**

["NO or VERY MINOR ethics concerns only"]

**Final Justification:**

The authors addressed my questions and concern, and thus I will maintain my score.

**Limitations:**

Yes

**Quality:**

4

**Strengths And Weaknesses:**

Strengths:
* The proof of this paper is sound. The authors provided a complete characterization of the hardness of calculating Nash equilibrium in two-team zero-sum games, and made progress towards understanding the hardness of calculating symmetric Nash equilibrium in general symmetric yet nonconvex-nonconcave setting.
* The results are important from game-theoretic perspectives.
* The reductions the authors provided are indeed original interesting.

Weaknesses:
* When overviewing the results in Section 1, it might be helpful to refer directly to the theorems in later sections. For example, I find the discussion around Theorem 1.5, as well its demonstration, to be a bit confusing; it might help if the authors could relate it more directly with Theorem 4.2. Furthermore, the author could provide a more clear contrast between their settings and the settings considered by Daskalakis et al. in 2021, when they discussed Theorem 4.4 in Section 1.

---

> ### Author Rebuttal · Authors · 2025-07-30
>
> We thank the reviewer for their valuable time and positive comments.
>
> > "When overviewing the results in Section 1, it might be helpful to refer directly to the theorems in later sections. Furthermore, the author could provide a more clear contrast between their settings and the settings considered by Daskalakis et al. in 2021, when they discussed Theorem 4.4 in Section 1."
>
> We thank the reviewer for the suggestion. In the revised version, we will spell out the relation between the statements in Section 1 and the subsequent statements.
>
> With regard to the comparison with Daskalakis et al. [1], as we highlight in Lines 298-299, the equilibrium notion we consider in our work is stronger compared to theirs. Specifically, if one takes a symmetric constraint $x \approx y$ in their notion, Player $x$ is only allowed to deviate to a specific set of strategies nearby $y$ (in order to satisfy the symmetry constraint), whereas our hardness results allow $x$ to unilaterally deviate to any other strategy. In other words, our reduction captures what Daskalakis et al. [1] refer to as “safe GDA fixed points,” which can be different from the usual GDA fixed points. We will highlight this difference in more detail in the revised version.
>
> Moreover, our work improves on the results of Daskalakis et al. in the following key ways: i) our reduction and proof are significantly simpler, making it easier for follow-up research to build on it; ii) our hardness result applies even when the accuracy parameter is constant, which is relevant in modern ML applications where one is often content with a crude solution quality; and iii) our class of functions comprises only simple quadratic functions, whereas the class of functions constructed by Daskalakis et al. is very involved.
>
> > "It is interesting that for symmetric games, all accurate Nash equilibria must be symmetric, whereas some approximate Nash equilibria are allowed to be slightly asymmetric. If some regulations are enforced on a symmetric game (say Lipschitz continuity of the payoff), can one understand a bit more about the hardness landscape?"
>
> Many of our reductions leverage the symmetry gadget we design (Line 223), which can be thought of as a “regularizer” that enforces symmetry. One limitation of that gadget is that it can only enforce symmetry between players in the *same* team. As the reviewer alludes to, it would be very interesting to create a gadget that enforces symmetry between players from *different* teams; this is indeed the crux to showing PPAD-hardness for min-max optimization, and would go a long way to settling the hardness landscape.
>
> ________
>
> References:
>
> [1] Constantinos Daskalakis, Stratis Skoulakis, and Manolis Zampetakis. The complexity of constrained min-max optimization. In Symposium on Theory of Computing (STOC), 2021.

---

> > ### Comment · Reviewer_1qji · 2025-08-04
> >
> > Thanks for the response. I acknowledge the rebuttal and would like to keep my current score.

---

### Official Review · Reviewer_BBrE · 2025-07-03

**Clarity:** 3
**Significance:** 3
**Originality:** 4
**Rating:** 5
**Confidence:** 1

**Summary:**

The paper makes significant progress in understanding the complexity of special classes of min-max optimization. In particular:

**Team zero-sum games (of which team adversarial games are a special case)** The authors show that this class of problems is CLS-complete by reducing to an earlier result from (Ghosh and Hollender, 2024) for symmetric identical-payoff 2-player games.

Moreover, they show that deciding whether an adversarial team game has an approximate unique equilibrium is NP-hard.

**Symmetric minmax games** They define the computational problem SymGDAFixedPoint and show that it is PPAD-Complete.
For the computational problem NonSymGDAFixedPoint, they show it is NP-hard.

**Questions:**

1. The paper is motivated with examples in robust ML, GAN training etc. Do the special classes of games considered directly apply to these settings? If not, what insights can extrapolate to practical settings?

2. How do the results extend to more than two teams, or games with overlap between teams?

**Ethical Concerns:**

["NO or VERY MINOR ethics concerns only"]

**Final Justification:**

The authors address my concerns in the rebuttal so I stick with my score.

**Limitations:**

Yes

**Quality:**

4

**Strengths And Weaknesses:**

## Strengths

**Significance** Understanding the complexity of min-max equilibria is both a very interesting problem from a theoretical perspective, and has relevance to the many places that such problems are now deployed in machine learning over the past few years.

**Originality** The reductions that the authors design to obtain their results is interesting, novel and presented clearly.

**Quality** The authors resolve questions about both small-constant adversarial team games and symmetric min-max problems, achieving tight CLS- and PPAD-completeness results where only partial bounds were known previously.

The authors provide a very comprehensive set of results, covering both existence (search) and uniqueness (decision) problems, two- and multi-team games, polymatrix and quadratic objectives, and both symmetric and non-symmetric equilibria, giving a unified view of the landscape.

## Weaknesses

**Clarity** The paper is quite difficult to read with each result presented twice -- first informally in the introduction and then in more detail in the main technical sections. I am not immediately sure how to address this, but it would be useful to make it more accessible to a broader NeurIPS audience

---

> ### Author Rebuttal · Authors · 2025-07-30
>
> We thank the reviewer for their valuable time and positive comments.
>
> > "The paper is quite difficult to read with each result presented twice -- first informally in the introduction and then in more detail in the main technical sections. I am not immediately sure how to address this, but it would be useful to make it more accessible to a broader NeurIPS audience"
>
> We provided informal versions of the main results in the introduction with the intention to make the paper more accessible and highlight the key takeaways without having to dive into the rigorous statement and technical details. We are happy to make any changes as the reviewer sees fit.
>
> > "The paper is motivated with examples in robust ML, GAN training etc. Do the special classes of games considered directly apply to these settings? If not, what insights can extrapolate to practical settings?"
>
> The main insight of our paper concerning such settings is that it is unlikely that there are general-purpose gradient-based algorithms that converge in a reasonable amount of time without making strong assumptions regarding the underlying optimization landscape. That being said, our hardness results do not rule out having efficient algorithms for more structured objectives and specific domains. For example, our current approach encompasses simplex or box constraints (hypercubes), but some problems are more naturally expressed in terms of $\ell_2$ ball constraints, such as GANs. Extending our hardness results to such settings is an interesting direction for future research.
>
> > "How do the results extend to more than two teams, or games with overlap between teams?"
>
> First, regarding games with more than two teams, three-player zero-sum games (that is, each team comprising a single player) are known to capture two-player general-sum games by adding a “dummy player.” So PPAD-hardness follows directly in that more general setting.
>
> When it comes to games with overlap between teams, we assume that the reviewer refers to situations where some players can change teams, either due to a strategic choice or because of some external stochasticity. This seems to us a very interesting direction for future research.

---

> > ### Comment · Reviewer_BBrE · 2025-08-05
> > **Response to Rebuttal**
> >
> > Thanks to the authors for their response and highlighting the interesting future work directions. I will keep my positive score.

---

### Official Review · Reviewer_3Gg9 · 2025-07-03

**Clarity:** 3
**Significance:** 3
**Originality:** 3
**Rating:** 5
**Confidence:** 2

**Summary:**

The paper considers several problems that arise in the context of
computing approximate Nash equilibria (or stationary points) in
games that involve two teams. Such problems are important in game
theory and they been studied by a number of researchers.  The current
paper settles the complexity of several problems that were left
open by previous work on the topic. The results make a nice
contribution to the area.

**Questions:**

(1) The footnote on page 3 mentions that symmetric zero-sum games are
ubiquitous in practical scenarios. It will be very helpful to
cite a few references to emphasize that point.

(2) Lines 125--132: In discussing the work of Bernasconi et al. [2024]
(who established Theorem 1.5 in the current paper independently)
you mention that your proof applies to simplex domains while their proof
applies to box domains. Can you please clarify the difference between the
two domains? In particular, is your proof applicable to a larger class of
problems compared to that of Bernasconi et al.?

(3) Theorem 1.7: You show this result for 6-player games (i.e., 3 in each team).
Are there any known complexity results for games with a smaller number of
players? Or is that an open problem as well?

A minor suggestion: In the bibliography, many references don't have complete
citations.  For example, many references are missing page numbers,
some arXiv papers don't have their arXiv IDs, etc. Since there is no limit
on the number of pages devoted to references, please include complete
citations.  Also, in some of the titles, words like "Nash" are not
capitalized. Please correct these in a future version of the paper.

**Ethical Concerns:**

["NO or VERY MINOR ethics concerns only"]

**Final Justification:**

I have read all the reviews, rebuttals and the subsequent discussion. I am glad to see that  the reviewers are unanimous  in recommending the acceptance of this paper. It s indeed an excellent contribution to the literature on game theory.

**Limitations:**

Yes

**Paper Formatting Concerns:**

None.

**Quality:**

4

**Strengths And Weaknesses:**

Strengths:

(a) The problems are of importance in game theory, a topic that
has assumed a lot of importance in AI. The paper provides a very good
discussion on the current state of the art regarding the problems
considered.

(b) The paper settles the complexity of several open problems on
the topic of approximate Nash equilibria for games involving teams.

(c) The proofs (inlcuded in the supplement) are quite non-trivial.

Weakness:

As far as this reviewer can tell, the paper doesn't appear to have any
major weakness. There are two (very) minor weaknesses due to the nature of
the results.

(a) The results are likely to be of interest to a limited group
of researchers.

(b) The paper provides a fair amount of intuition regarding the results
and constructions. Despite this, readers not familiar with the technical
ideas may find the paper somewhat difficult to understand.

---

> ### Author Rebuttal · Authors · 2025-07-30
>
> We thank the reviewer for their valuable time and positive comments.
>
> > "The results are likely to be of interest to a limited group of researchers."
>
> Although some of the techniques we employ are specialized, we believe that the main takeaways of our results should be of interest quite broadly to the NeurIPS community. Many central problems in machine learning and reinforcement learning can be cast as min-max optimization problems, such as adversarial training and GANs. Our results contribute to understanding the fundamental barriers concerning min-max optimization problems and adversarial team games, and provide an explanation for why natural gradient-based algorithms often fail to stabilize in nonconvex-nonconcave settings.
>
> > "The paper provides a fair amount of intuition regarding the results and constructions. Despite this, readers not familiar with the technical ideas may find the paper somewhat difficult to understand."
>
> We thank the reviewer for this feedback. We will provide further details concerning the key technical ideas in the main body in order to make the paper more accessible in the revised version.
>
> > "The footnote on page 3 mentions that symmetric zero-sum games are ubiquitous in practical scenarios. It will be very helpful to cite a few references to emphasize that point."
>
> Many popular recreation games commonly used for AI benchmarking, such as poker and battleship, are symmetric games; the symmetry assumption is especially natural since it guarantees that no player has an a priori advantage before the game begins. Also, the most common examples in game theory, such as rock-paper-scissors and matching pennies, are symmetric; such games were already present in the original work of Von Neumann [1, p.303]. We will include this discussion in the revised version.
>
> > "Lines 125--132: In discussing the work of Bernasconi et al. [2024] (who established Theorem 1.5 in the current paper independently) you mention that your proof applies to simplex domains while their proof applies to box domains. Can you please clarify the difference between the two domains? In particular, is your proof applicable to a larger class of problems compared to that of Bernasconi et al.?"
>
> The starting point of our reduction and the resulting hard class of min-max optimization problems is different from that considered by Bernasconi et al., both in terms of the domain and the underlying objective function. It is important to note that the domain crucially affects the complexity of the problem. For box constraints, solving linear VIs with even a constant accuracy is PPAD-hard [2]. On the other hand, for constant accuracy and under simplex constraints, there is a well-known quasipolynomial-time algorithm [3]; one needs to have inverse polynomial accuracy to get PPAD-hardness under simplex constraints [4]. Besides the difference in the domain, our class of problems is symmetric, which enables us to prove hardness for computing symmetric equilibria.
>
> > "Theorem 1.7: You show this result for 6-player games (i.e., 3 in each team). Are there any known complexity results for games with a smaller number of players? Or is that an open problem as well?"
>
> The basic reason we are only able to obtain hardness results for 3 vs. 3 (instead of 2 vs. 2) team zero-sum games is that the symmetry gadget we introduce (Line 223) requires two players ($x, y$) on one team and another player ($z$) to be on the other team. As a result, in team zero-sum games, since we need to enforce symmetry in both teams, a total of 6 players is the best we can achieve. It is an interesting open problem to see if we can extend the hardness result to 2 vs. 2 team games, but we believe this would require additional technical ideas.
>
> > "In the bibliography, many references don't have complete citations. Also, in some of the titles, words like "Nash" are not capitalized. Please correct these in a future version of the paper."
>
> We thank the reviewer for pointing this out. We will correct the inconsistencies in the bibliography in the revised version.
>
> ________
>
> References:
>
> [1]  Von Neumann J, Zur Theorie der Gesellschaftsspiele. Mathematische Annalen 100:295–320, 1928.
>
> [2] Aviad Rubinstein. Inapproximability of Nash equilibrium. In Proceedings of the fortyseventh annual ACM symposium on Theory of computing, pp. 409–418, 2015.
>
> [3] Richard J Lipton, Evangelos Markakis, and Aranyak Mehta. Playing large games using simple strategies. In Proceedings of the 4th ACM conference on Electronic commerce, pages 36–41, 2003.
>
> [4] Xi Chen, Xiaotie Deng, and Shang-Hua Teng. Settling the complexity of computing two-player Nash equilibria. J. ACM, 56(3):14:1–14:57, 2009.

---

> > ### Comment · Reviewer_3Gg9 · 2025-08-05
> > **Thanks for the detailed responses**
> >
> > I have gone through all the reviews and the responses by the authors. I thank the authors for going through my comments and providing careful responses. These responses nicely  address all my concerns. I believe that the paper represents a nice contribution to game theory.  I am glad to recommend its acceptance.

---

### Decision · Program_Chairs · 2025-09-17

**Decision:**

Accept (spotlight)

**Comment:**

This work investigates the computational complexity of equilibrium computation in adversarial team games. Compared to related works such as Daskalakis et al, the authors develop a reduction technique to provide some new contributions; in particular, the class of functions they use comprises only quadratic functions, which is potentially modular and helps lead to a simpler proof. Their results include the computational hardness result of computing equilibria in 3-player adversarial team games (2 versus 1) and that of Nash equilibria computation in symmetric n-dimensional min-max optimization. For the latter, the authors proved PPAD-completeness through polynomial-time reduction from symmetric Nash equilibrium calculation in two-player symmetric games, which is known to be PPAD-complete.

All reviewers unanimously support accepting this paper, and hence it is put into the acceptance category. Still, I encourage the authors to incorporate some of the feedback from the reviewers (e.g., the ones by Reviewer 1qji and JDAf) and their responses during the rebuttal phase into the next version.